# Membraneless channels sieve cations in ammonia-oxidizing marine archaea

Andriko von Kügelgen[1,2], C. Keith Cassidy[3], Sofie van Dorst[2], Lennart L. Pagani[2], Christopher Batters[4], Zephyr Ford[2], Jan Löwe[1], Vikram Alva[5], Phillip J. Stansfeld[6] & Tanmay A. M. Bharat[1✉]

*Nitrosopumilus maritimus* is an ammonia-oxidizing archaeon that is crucial to the global nitrogen cycle[1,2]. A critical step for nitrogen oxidation is the entrapment of ammonium ions from a dilute marine environment at the cell surface and their subsequent channelling to the cell membrane of *N. maritimus*. Here we elucidate the structure of the molecular machinery responsible for this process, comprising the surface layer (S-layer), using electron cryotomography and subtomogram averaging from cells. We supplemented our in situ structure of the ammonium-binding S-layer array with a single-particle electron cryomicroscopy structure, revealing detailed features of this immunoglobulin-rich and glycan-decorated S-layer. Biochemical analyses showed strong ammonium binding by the cell surface, which was lost after S-layer disassembly. Sensitive bioinformatic analyses identified similar S-layers in many ammonia-oxidizing archaea, with conserved sequence and structural characteristics. Moreover, molecular simulations and structure determination of ammonium-enriched specimens enabled us to examine the cation-binding properties of the S-layer, revealing how it concentrates ammonium ions on its cell-facing side, effectively acting as a multichannel sieve on the cell membrane. This in situ structural study illuminates the biogeochemically essential process of ammonium binding and channelling, common to many marine microorganisms that are fundamental to the nitrogen cycle.

The ocean is our planet's largest biome, where pelagic microbial Thaumarchaeota (syn. Nitrososphaerota) represent one of the most abundant organisms[3]. The numerical dominance of marine thaumarchaea suggests that they have a major role in global biogeochemical cycles[1,2]. *N. maritimus*, an intensely studied marine thaumarchaeon, grows chemolithoautotrophically by aerobically oxidizing ammonia to nitrite[1]. This organism has also been shown to regenerate oxygen under anoxic conditions, and to fix $CO_2$[4,5], placing it in an important position in the global nitrogen and carbon dioxide biogeochemical cycles.

Owing to the low concentration of ammonia ($NH_3$) or ammonium ($NH_4^+$) ions in the oceans, reported in the $10^{-8}$ to $10^{-9}$ M range[6], marine archaea such as *N. maritimus* have evolved specialized molecular machinery to attract ammonium ions on their cell surface to facilitate ammonium oxidation at the cell membrane[7]. *N. maritimus*, like most archaeal cells, is encased by a paracrystalline, proteinaceous surface layer or S-layer[8–11]. Bulk modelling of the *N. maritimus* S-layer has predicted that this cellular organelle might have a key role in elevating the ammonium concentrations in the pseudoperiplasmic space between the S-layer and the cell membrane[12,13]; however, how this occurs on a mechanistic and molecular level is unclear. At the overall morphological scale, the *N. maritimus* S-layer has been reported to have a hexagonal arrangement[9,12] and is postulated to consist of repeating subunits of the proteins Nmar_1547 or Nmar_1201 (two proteins with 91% sequence identity), based on transcriptomic and proteomic data[14,15]. Here, to understand the rules governing ammonium binding and enrichment by *N. maritimus* and related marine Thaumarchaeota, we investigated the molecular structure of the *N. maritimus* S-layer using structural, biochemical, cellular and bioinformatic methods.

## Cryo-ET analysis of the *N. maritimus* S-layer

To gain insights into this problem, we used electron cryotomography (cryo-ET) and subtomogram averaging techniques. We have previously applied these methods to determine in situ structures of prokaryotic S-layers from in vitro purified specimens containing cellular fragments[16–18]. Our goal was to solve the structure of the *N. maritimus* S-layer directly from whole cells. Cryo-ET analysis of *N. maritimus* cells revealed an S-layer surrounding the approximately 300-nm-wide elongated cells, which contained a compact nucleoid and several cytosolic ribosomes (Fig. 1a). Using subtomogram averaging, we mapped the locations of the S-layer repeating units on the cell, which were arranged in a hexagonal array (Fig. 1b). We found that the S-layer hexamers coated the cells with near-perfect continuity along their lengths, while local pentameric defects were present on the cell edges, completing the S-layer (Fig. 1b), which was confirmed by quantification of hexamer and pentamer positions relative to the centre of the cell (Fig. 1c).

[1]Structural Studies Division, MRC Laboratory of Molecular Biology, Cambridge, UK. [2]Sir William Dunn School of Pathology, University of Oxford, Oxford, UK. [3]Department of Physics and Astronomy, University of Missouri-Columbia, Columbia, MO, USA. [4]Protein and Nucleic Acid Chemistry Division, MRC Laboratory of Molecular Biology, Cambridge, UK. [5]Department of Protein Evolution, Max Planck Institute for Biology Tübingen, Tübingen, Germany. [6]School of Life Sciences and Department of Chemistry, University of Warwick, Coventry, UK. ✉e-mail: tbharat@mrc-lmb.cam.ac.uk

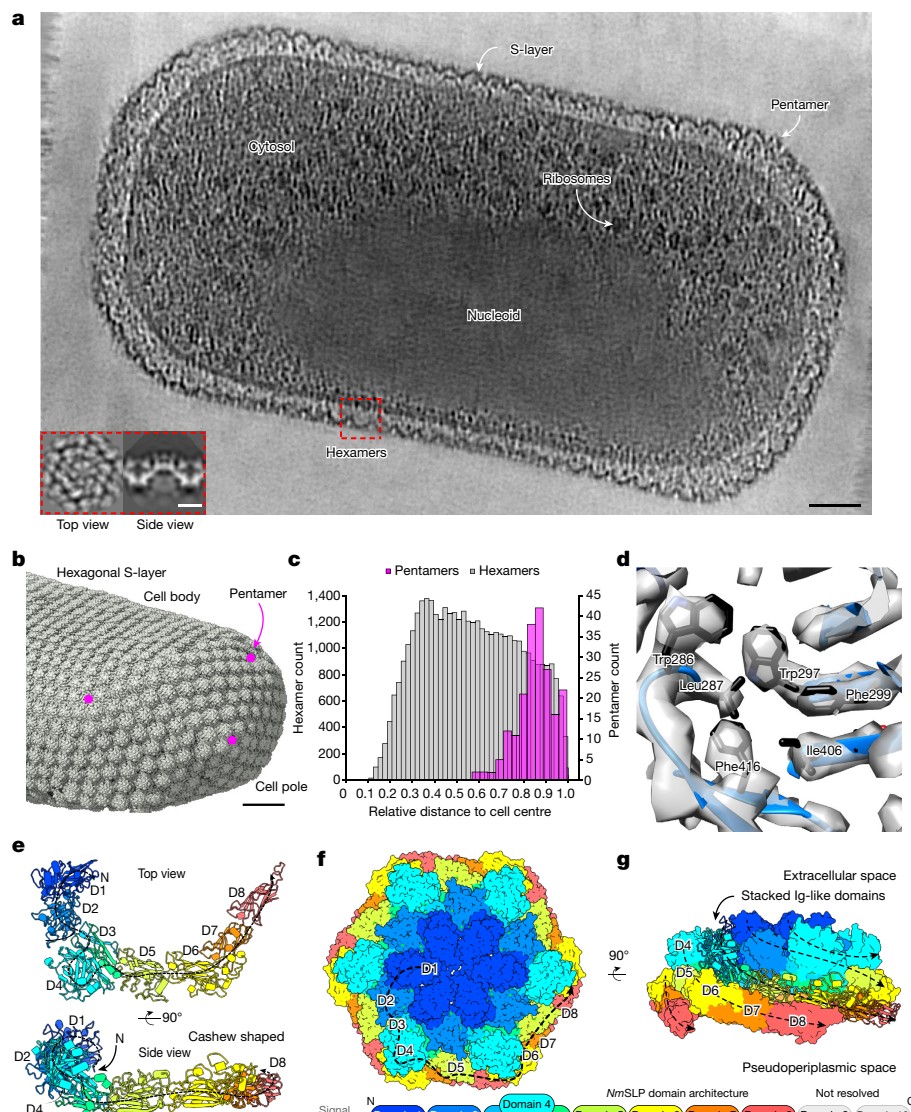

**Fig. 1 | The molecular structure and assembly of the *N. maritimus* S-layer in intact cells. a**, A denoised[36,37] tomographic slice through a *N. maritimus* cell shows ultrastructural details of this marine archaeon (annotated). Inset: top and side views of the subtomogram average of the S-layer. Scale bars, 500 Å (main image) and 100 Å (inset). Cellular tomography was performed at least 27 times (Extended Data Table 1). **b**, Map of the subtomogram positions in the cellular S-layer showing the presence of pentameric defects at the edge of the cell. Scale bar, 500 Å. **c**, A histogram of subtomogram positions from all tomograms relative to the three-dimensional centre of the cell body (*n* = 41,303 hexamers (grey) and *n* = 203 pentamers (pink); one out of two biological replicates shown). **d**, The subtomogram averaging map enables derivation of a molecular model directly from cellular data. Amino acid residue side chains resolved are marked. See also Extended Data Figs. 1 and 2. **e**, A ribbon model of the cashew-shaped *Nm*SLP monomer is shown in two orthogonal views. **f,g**, The structure of the S-layer hexamer displayed in two orthogonal views shows that *Nm*SLP monomers are arranged as an array of Ig-like domains; each domain (D) is coloured differently (a schematic is shown below). The first eight Ig-like domains are resolved in the cryo-ET and subtomogram averaging map.

Next, we used state-of-the-art cryo-ET imaging and image-processing workflows, which have been shown to support high-resolution in situ structure determination from purified specimens[19]. As a result, we produced a high-resolution map of the *N. maritimus* S-layer hexamer from intact cells (Fig. 1d, Extended Data Fig. 1, Extended Data Table 1 and Supplementary Videos 1 and 2). The central region of the S-layer hexamer in the map had a resolution of 3.3 Å, with the resolution decaying to around 4.5 Å towards the periphery (Extended Data Fig. 1). The subtomogram averaging map contained sufficient details to enable us to derive an atomic model of the S-layer (Fig. 1d–g and Extended Data Fig. 2). The structure shows that the S-layer is pseudohexagonal (Fig. 1f,g) and consists of the repeated interactions of the Nmar_1547 (hereafter, *Nm*SLP) S-layer protein. Despite the high sequence similarity of *Nm*SLP to the other previously predicted SLP, Nmar_1201, a unique segment of *Nm*SLP between residues 911 and 977 was clearly resolved

in our map. This enabled us to identify *Nm*SLP as the primary *N. maritimus* SLP on cells through direct structure determination (Extended Data Fig. 2). This observation confirms previous transcriptomic data showing high expression levels of the *Nmar_1547* gene compared with *Nmar_1201*[14]. However, we cannot rule out that Nmar_1201 could be present at lower copy numbers.

At the sequence level, *Nm*SLP is arranged into ten immunoglobulin-like (Ig-like) domains (Fig. 1f,g). The first eight domains were well resolved in our 3.3–4.5-Å-resolution subtomogram averaging map. By contrast, the last two domains appeared less ordered, with the local resolution decaying towards the C terminus of *Nm*SLP, away from the centre of the S-layer hexamer (Extended Data Figs. 1 and 2). At the N terminus, the $C_6$ symmetry of the hexamer is broken, revealing a distinctly two-fold symmetric central pore (Extended Data Fig. 2). Each monomer of *Nm*SLP in the S-layer adopts a rough 'cashew' shape, facilitating

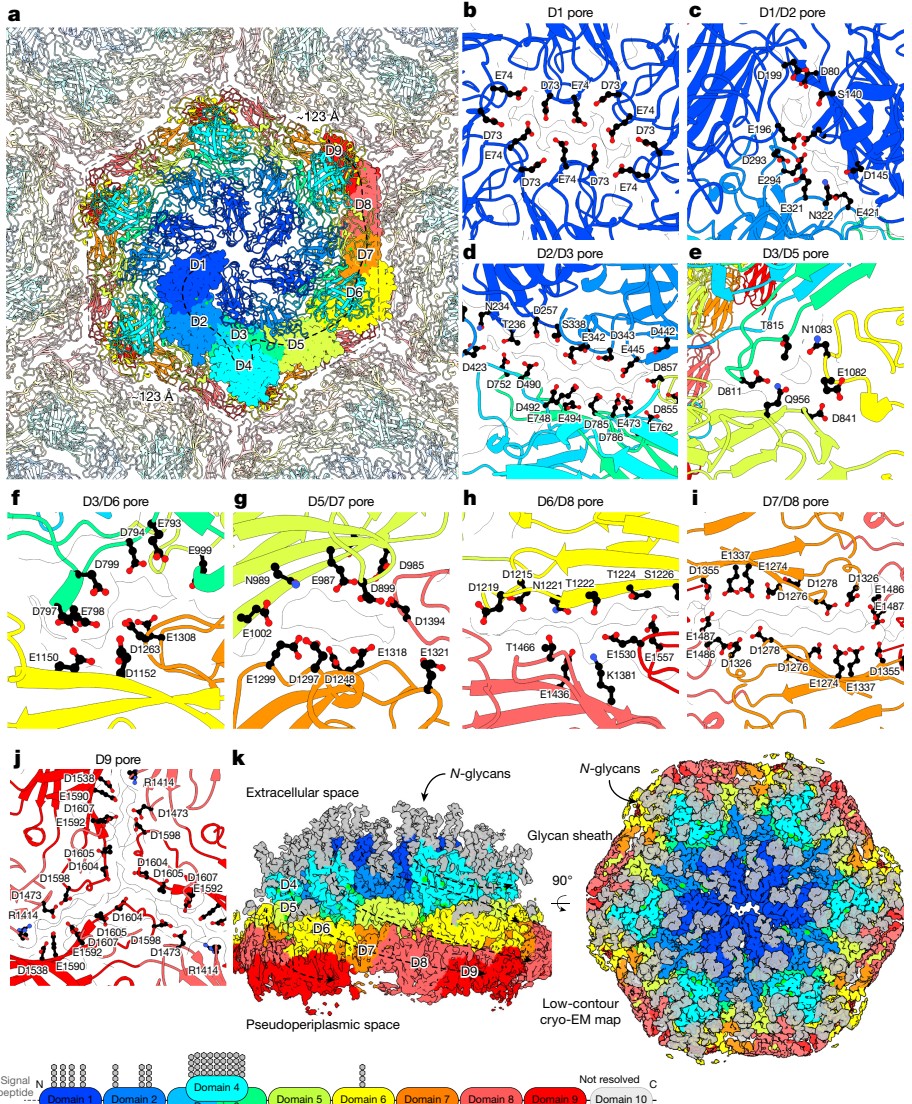

**Fig. 2 | Cryo-EM structure of isolated *N. maritimus* S-layer sheets. a**, In vitro cryo-EM structure (global resolution, 2.7 Å) of isolated S-layer sheets from *N. maritimus*. The colour scheme for ribbon diagrams is the same as in Fig. 1; domains of one *Nm*SLP are marked. **b–j**, Magnified views of the pores lined with negatively charged residues, which are ubiquitous in the S-layer sheet. The location of the pores is given in the titles of the panels. **k**, The sharpened[38] cryo-EM map shows 17 glycans decorating each *Nm*SLP; the map shown at a lower contour level in two different orientations. A schematic of the glycan locations on the *Nm*SLP sequence is shown below.

several interactions around the hexamer of *Nm*SLP (Fig. 1e–g). Domain 4, nestled within domain 3 and linked by short connectors, is slightly raised relative to the base of the cashew-shaped Ig-array of *Nm*SLP, projecting towards the extracellular milieu.

Although the amino acid residues (37–1499) from the first eight Ig-like domains could be unambiguously built into the subtomogram averaging map, several unexplained densities were observed, emanating from surface-exposed asparagine residues (Extended Data Fig. 2). Given that archaeal SLPs are known to be heavily glycosylated[20], we hypothesized that these densities might correspond to glycans. Another notable set of unexplained densities was observed near the negatively charged amino acid residue side chains of Asp73 and Glu74 at the central $C_2$ pore, as well as between *Nm*SLP monomers around the hexamer (Extended Data Fig. 2). Considering that positively charged ions have been previously observed bound to S-layers[21,22], we hypothesized that these additional densities on the *N. maritimus* S-layer could potentially correspond to bound cations, although the resolution of the map prevented us from unambiguously assigning their chemical identities.

## Cryo-EM shows a porous S-layer

The *N. maritimus* S-layer has previously been predicted through bulk biophysical modelling to help attract ammonium ions[12]. Considering the tight sheath formed by the S-layer around cells (Fig. 1b,c), providing an extremely large surface area for interaction with the marine environment, we postulated that the S-layer functions as a negatively charged ammonium trap. This would facilitate the movement of cations such as ammonium towards the cell membrane, specifically to the sites of ammonium oxidation. To test these hypotheses with higher resolution structures, in which ion and other densities would be better resolved, we purified *N. maritimus* cell envelopes for cryo-electron microscopy (cryo-EM) analysis. We then used single-particle techniques, as applied previously to two-dimensional S-layer sheets for structure determination[18,23], to solve a 2.7-Å-resolution structure of the S-layer (Fig. 2a, Extended Data Fig. 3 and Extended Data Table 2).

The single-particle structure was very similar (root mean squared deviation of 2.21 Å for the full composite model, 0.54 Å for residues 35–455 (refined in $C_2$) and 1.39 Å for residues 466–1498 (refined in $C_6$))

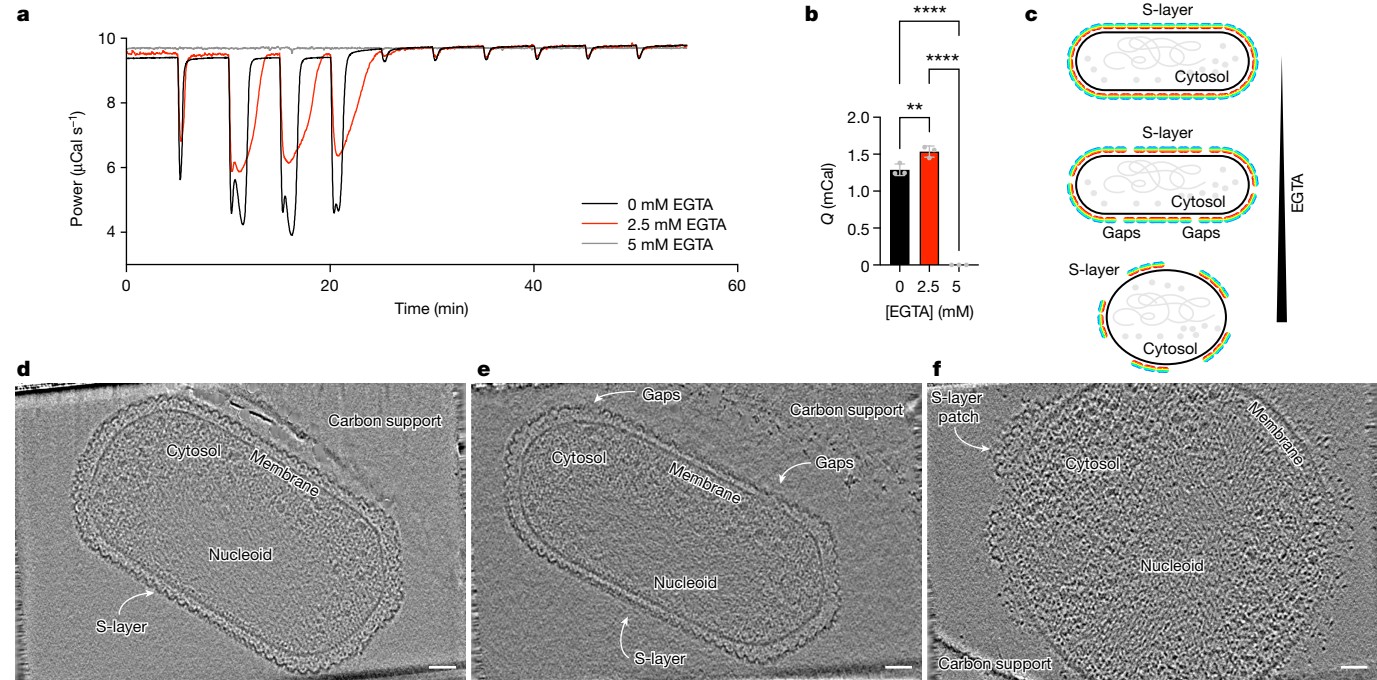

**Fig. 3 | Ammonium ions either bind directly to the S-layer or pass through it to bind to the underlying cell. a**, ITC signal of intact *N. maritimus* cells titrated with ammonium chloride with different pretreatments with EGTA. **b**, Quantification of the ITC curves showing the total heat (Q) released in the different experiments. Data are mean ± s.d. $n = 3$ biologically independent replicates. Statistical analysis was performed using ordinary one-way analysis of variance (ANOVA) with correction for multiple comparisons; ****$P < 0.0001$, **$P = 0.0075$. **c**, Schematic of the ITC and cryo-ET experiment presented.

**d**, Cryo-ET analysis of *N. maritimus* cells after titration with ammonium chloride shows complete coating with an S-layer. $n = 150$ from two biological replicates. **e**, Cryo-ET analysis of *N. maritimus* cells treated with 2.5 mM EGTA after titration with ammonium chloride shows gaps in the S-layer in a heterogeneous cell population. $n = 83$ from two biological replicates. **f**, Cryo-ET analysis of *N. maritimus* cells treated with 5 mM EGTA after titration with ammonium chloride showing round cells with a naked membrane. $n = 152$ from three biological replicates. For **d**–**f**, scale bars, 500 Å.

to the subtomogram averaging structure (Fig. 2, Extended Data Fig. 4 and Supplementary Video 3), enabling us to extend our structure by modelling the ninth Ig-like domain, reaching up to residue 1616 out of 1734 (Fig. 2a and Extended Data Fig. 3). The last Ig-like domain remains unresolved in our map, with only disordered, diffuse density observed beyond the ninth Ig-like domain of *Nm*SLP in the pseudo-periplasmic space, indicating flexibility relative to the rigid part of the S-layer. The *Nm*SLP hexamer in the single-particle structure appears to be slightly expanded compared with the subtomogram averaging structure (Supplementary Video 3), perhaps due to differences in the S-layer curvature.

The *N. maritimus* genome also encodes a homologue for a cell-anchoring SlaB protein (Extended Data Fig. 5a,b and Supplementary Table 1) that is known to bind the S-layer of the archaeon *Sulfolobus acidocaldarius* to the cell membrane[24]. However, proteomic data on *N. maritimus* indicate that this protein is considerably less abundant in the cell than *Nm*SLP[15]. We speculate that the last (tenth), unresolved Ig-like domain of *Nm*SLP may have a role in anchoring the S-layer to molecules present on the cell envelope, therefore partially reducing the need for stoichiometric anchoring by a SlaB or a SlaB-like protein in *N. maritimus*.

In our single-particle structure, the subunit contacts between *Nm*SLP hexamers (Fig. 2b–j) were better resolved compared with in our subtomogram averaging map (Fig. 1). Notably, several contact sites, both between hexamers and within each hexamer, contain pores lined predominantly by rows of negatively charged amino acid residues (Fig. 2b–j). These pores are relatively small (around 5 Å) but are compatible with the size of small chemical species, fitting with the idea that the pores of the S-layer may function as cation channels. Owing to the repeating pattern of the S-layer, these pores span the entirety of the *N. maritimus* cell surface. Supporting the anticipated cation-binding

properties of the S-layer, a bioinformatics comparison of the amino acid composition of *Nm*SLP with that of the *N. maritimus* proteome and all archaeal proteins revealed a substantial increase in aspartic acid residues (10.9% in *Nm*SLP versus 6.1% in the *N. maritimus* proteome versus 7.3% in all archaea). Concurrent with this, the percentages of lysine and arginine residues are reduced (1.7% and 2.2% in *Nm*SLP versus 8.4% and 3.4% in the *N. maritimus* proteome versus 3.8% and 5.9% in all archaea), resulting in an S-layer that is highly negatively charged.

Overall, our structure shows that the *Nm*SLP monomers densely populate the S-layer sheet. This arrangement is reminiscent of the S-layer in the Dead Sea archaeon *Haloferax volcanii*, which is composed of an SLP called csg, which also consists of tandemly repeated Ig-like domains[18]. Such arrays of Ig-like domains have been observed in archaeal[18,25], monoderm bacterial[22,26] and diderm bacterial S-layers[23,27]. Although these SLPs share some structural similarities, they diverge notably at the sequence level, as well as at the overall organizational level containing different number of domains (Extended Data Fig. 5c), enabling them to assemble into unique two-dimensional sheets[23,24], each with distinctly different glycosylation patterns and cell anchoring mechanisms.

In total, 17 glycan densities per monomer of *Nm*SLP were also resolved in the single-particle map (Fig. 2k and Extended Data Fig. 4). Although these densities do not support direct derivation of the chemical structure of the glycans, they project away from the cell surface at asparagine residues, which are followed by a threonine or serine residue at the +2 position (Fig. 2k). Enshrouding the outer domains of the proteinaceous *Nm*SLP S-layer, these glycans form a thick shell, encasing the cell in a sugar-rich coat (Fig. 2k). The mesh-like arrangement of the glycans probably provides protection, potentially shielding the cell from phages[28]. It might also enhance the hydrophilicity of the cell surface, making it suitable for marine environments. Most of these glycans are

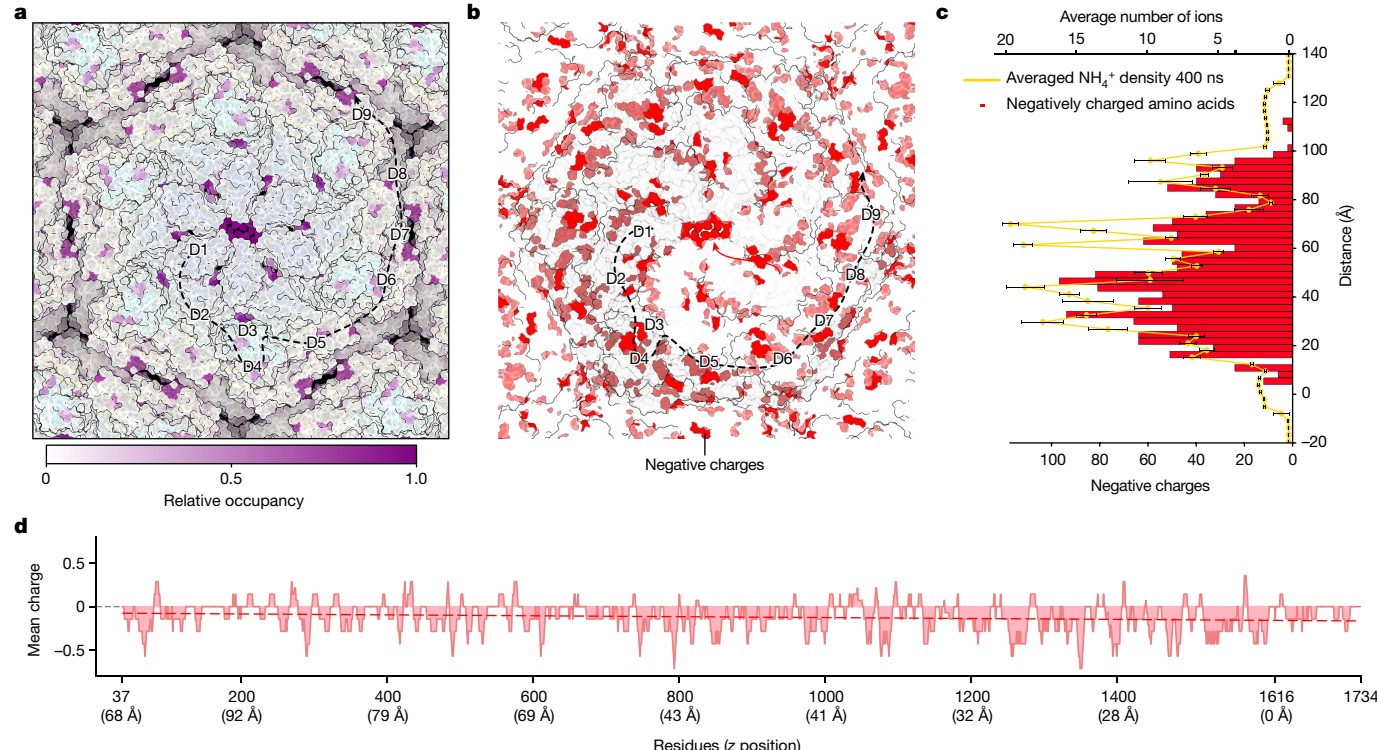

**Fig. 4 | Ammonium (NH₄⁺) binding to the negatively charged S-layer lattice.**
**a**, MD simulations support ammonium-ion binding at the S-layer pores. Residue-based ammonium occupancies during the 0.1 M NH₄⁺ MD simulations are plotted onto the S-layer structure on a relative scale from white to purple.
**b**, The distribution of negatively charged residues (shown in red) in the S-layer matches well with the MD simulations showing predicted ammonium-binding residues (Extended Data Fig. 6). **c**, A histogram of negatively charged residues along the S-layer, overlaid onto the ammonium-ion positions in the 0.1 M NH₄⁺ MD simulations (distance was calculated from the closest, membrane-proximal amino acid residue in the S-layer structure). For the averaged ammonium-ion residence from three independent MD simulations (averaged over the last 400 ns of each simulation), the error bars denote ±1 s.d. (Supplementary Fig. 2). **d**, The mean local charge of *Nm*SLP plotted along the sequence shows a gradual but continual increase in negative charge. The *z* position of the *Nm*SLP residues, derived from the S-layer structure, is indicated, with the ninth Ig-like domain forming the base of the S-layer, proximal to the cell membrane.

located in the N-terminal segment of *Nm*SLP, primarily in outermost domains 1, 2 and 4, with a single glycan present in domain 6 (Fig. 2k).

## Ammonium binding of the S-layer

We confirmed that the concentration of ammonium ions in the medium strongly influences the growth of *N. maritimus* (Supplementary Fig. 1a), as shown previously[4,9]. To directly measure ammonium ion binding to the cell surface, we performed isothermal titration calorimetry (ITC), titrating a medium containing NH₄Cl against whole *N. maritimus* cells. As expected, growing *N. maritimus* cells showed strong and robust ammonium binding (Fig. 3a,b and Supplementary Fig. 1b; *n* = 3). We subsequently performed cryo-ET analysis of the same sample after ITC measurements, revealing normal cell morphology with an intact S-layer coating the cells (Fig. 3c,d; *n* = 150). As ammonium is the sole energy source for these growing cells, ammonium ions must somehow reach the cell for oxidation; we therefore inferred that the measured ammonium binding occurs either directly to the S-layer or to the underlying cell after passage through the S-layer (Fig. 3c,d). We then perturbed the S-layer by pretreating the cells with ethylene glycol-bis(β-aminoethyl ether)-*N*,*N*,*N'*,*N'*-tetraacetic acid (EGTA), which is known to impair several prokaryotic S-layers[16]. *N. maritimus* cells that were pretreated with 2.5 mM EGTA showed altered ammonium binding (Fig. 3a,b and Supplementary Fig. 1b; *n* = 3), concurrent with observed gaps and partial S-layer disruptions seen in cryo-ET in a heterogeneous population (Fig. 3e; *n* = 83). Near-complete disruption of the S-layer with 5 mM EGTA entirely abolished ammonium binding (Fig. 3a,b and Supplementary Fig. 1b; *n* = 3), leading to rounding up

of cells with exposed, uncoated membranes (Fig. 3f; *n* = 152). These experiments indicate that an intact S-layer is critical for ammonium binding and may also be important for cell shape maintenance in *N. maritimus* in a calcium-dependent manner.

To examine the biochemically observed ammonium-binding properties of the S-layer structurally, we purified S-layer sheets in vitro, at a higher concentration of ammonium (2.5 mM compared to 1 mM NH₄Cl) and resolved a 3.1-Å-resolution cryo-EM structure of the S-layer enriched in ammonium ions. Compared with our original cryo-ET structure (Supplementary Fig. 1c–f), this ammonium-enriched S-layer structure showed an increased number of unexplained densities at several S-layer pores, indicative of ammonium ion binding to the negatively charged amino acid residues lining these pores (Supplementary Fig. 1c–f).

To further investigate the ability of the lattice to bind to cations, we performed atomistic molecular dynamics (MD) simulations of the *N. maritimus* S-layer in the presence of ammonium (Fig. 4a, Methods, Extended Data Fig. 6 and Supplementary Fig. 2), using a framework that has recently been shown to yield strong agreement with X-ray crystallography and cryo-EM studies of cation binding to S-layers[21]. Analysis of residue-based ammonium occupancies over the course of three 500 ns simulations using PyLipID[29] clearly identified the primary acidic residues that mediate ammonium binding (Supplementary Table 2 and Supplementary Video 4). Moreover, mapping the high-ammonium occupancy (>50%) residues onto the hexamer structure revealed multiple acidic residue clusters in excellent agreement with the pores identified in our cryo-EM and cryo-ET maps (Fig. 4b and Extended Data Fig. 7). These observations are compatible with the idea that the

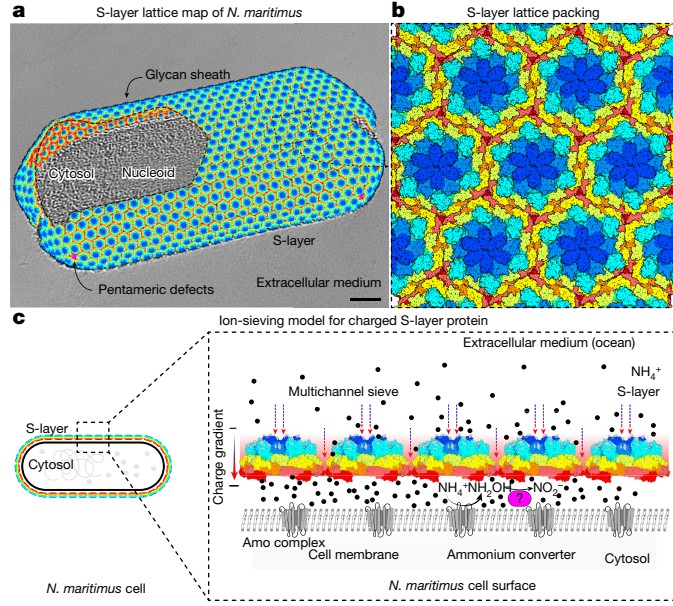

**a** S-layer lattice map of *N. maritimus*

Glycan sheath

Cytosol  Nucleoid

S-layer

Pentameric defects

Extracellular medium

**b** S-layer lattice packing

**c** Ion-sieving model for charged S-layer protein

Extracellular medium (ocean)

NH₄⁺
S-layer

Multichannel sieve

Charge gradient

S-layer

Cytosol

*N. maritimus* cell

NH₄⁺ NH₂OH NO₂⁻

Amo complex

Cell membrane  Ammonium converter  Cytosol

*N. maritimus* cell surface

**Fig. 5 | The in situ structure of the *N. maritimus* S-layer shows a multichannel sieve nearly perfectly coating cells at high-copy numbers. a**, The S-layer lattice coats nearly the entire outer surface of *N. maritimus* cells. A small part of the lattice map (black dashed line) has been cut out from the top of the cell for clarity. The pseudohexagonal lattice is joined together by pentameric defects (Extended Data Fig. 8). Scale bar, 500 Å. Cellular tomography was performed at least 32 times (Extended Data Table 1). **b**, A model of the S-layer lattice is shown as a space-filling representation with the same colour scheme as in Fig. 1. **c**, The ion-exchanging model for the *N. maritimus* S-layer. The highly negatively charged S-layer entraps ammonium ions, and these ions may move along multiple pores in the lattice, aided by increasing negative charge along the lattice. Once beyond the S-layer, the ammonium ions can diffuse to the membrane predominantly composed of crenarchaeol[39], where they are converted to nitrite through hydroxylamine ($NH_2OH$) as an intermediate by an unknown protein complex (pink)[34], setting up an ion sink and concentration gradient between the S-layer and the membrane.

multiple pores in the S-layer sheet, lined with negatively charged residues, facilitate the movement of ammonium ions across the S-layer. Furthermore, in our simulations, we also found that ammonium ions could be replaced with sodium ions, suggesting that the S-layer might not discriminate between different small positively charged ions for binding (Supplementary Fig. 2).

Collectively, our structural data reveal that the S-layer of *N. maritimus* functions as a multichannel exchanger for ammonium ions, featuring negatively charged residues that line several pores in the S-layer (Fig. 2b–j). In this context, it is interesting that the negative charge of the S-layer increases when moving from the extracellular environment toward the cell membrane (Fig. 4c), specifically from the N to the C terminus of each *Nm*SLP. This charge gradient probably facilitates the movement of ammonium ions through the S-layer towards the cell membrane. Our MD simulations support this hypothesis, showing an accumulation of ammonium ions at the cell-facing side of the S-layer (Fig. 4c, Supplementary Fig. 2 and Supplementary Video 4). To ensure that these observations did not depend on the amount of ammonium present, we conducted further MD simulations of the S-layer system in 0.05 M or 0.2 M ammonium, reproducing in both cases strong ammonium binding by the S-layer and relative accumulation of ammonium ions towards the cell-facing side of the S-layer (Supplementary Fig. 2).

The observed increase in negative charge at the structural level is also mirrored at the sequence level. There is a continuous and substantial increase in negative charge from the N to the C terminus of

the *Nm*SLP protein sequence (Fig. 4d), with the membrane-proximal C terminus being highly negatively charged. Using sensitive sequence-based homology searches and structure prediction, we found that Ig-domain-containing SLPs with a charge gradient are common across all described groups of ammonia-oxidizing archaea (AOA), suggesting similar S-layer arrangements in these species (Extended Data Fig. 7). Although the overall domain organization of such AOA and other archaeal SLPs could be similar (Extended Data Fig. 5a,b), the number of Ig-like domains and the charge distribution in each SLP vary, possibly reflecting differences in the function of these S-layers in binding positively charged molecules and ions (Extended Data Fig. 7d).

## Molecular modelling of the cell surface

Together, our data enabled us to construct a molecular model of the *N. maritimus* cellular S-layer (Fig. 5a,b). Subtomogram position mapping of the S-layer hexamers and pentamers demonstrates that the S-layer coats the cell surface with near-perfect continuity (Fig. 5a,b), a characteristic that is also observed in other archaeal S-layers[18]. The continuous S-layer is heavily glycosylated, possibly protecting the cell from phages in the harsh marine environment. The dome-shaped structure of the S-layer hexamer probably supports flexibility, allowing *Nm*SLP to coat different parts of the cell membrane with varied curvature. The S-layer is closed around cells by pentameric positions, which also appear to be composed of *Nm*SLP (Extended Data Fig. 8). At the technical level, this study highlights the power of modern structural biology—molecular structures obtained from whole-cell cryo-ET, in conjunction with in-cell biochemistry and MD simulations, can provide key biochemical and mechanistic insights. With advancements in data collection and image-processing methodologies[19,30,31], we anticipate that such in situ structural techniques will significantly enhance our molecular understanding of cells (Supplementary Fig. 3).

## Conclusion

Our data are consistent with a scenario in which ammonium and other cations are bound and enriched at the cell surface by the negatively charged S-layer (Fig. 5c). The S-layer provides an extremely large surface area for interaction with the surrounding marine environment, where it acts as a multichannel cation exchanger and, due to its gradually increasing negative charge, leads to the accumulation of ammonium ions on the cell-facing side of the S-layer. An 'ammonium sink' exists at the cell membrane, where the integral membrane machinery, ammonia monooxygenase[32,33], converts $NH_4^+$ to $NO_2^-$ with hydroxylamine ($NH_2OH$) as an intermediate[34]. This activity probably establishes an ammonium concentration gradient, extending from the membrane-proximal, ammonium-rich side of the S-layer to areas of lower concentrations near the cell membrane. A previous biophysical model has outlined how such a sink could function[12], potentially initiating a chain of ammonium uptake that provides energy to the cell (Fig. 5c). This passive charge gradient of the S-layer could explain why marine AOA exhibit a 200-fold greater affinity for ammonia compared with ammonia-oxidizing bacteria[7], which generally lack S-layers. This increased affinity enables AOA to thrive in low-ammonia marine environments. The energy-efficient, passive enrichment mechanism is particularly advantageous in the resource-limited, harsh conditions that are found in the ocean depths. Moreover, as S-layers similar to that of *N. maritimus* are found in nearly all AOA, including those in soil ecosystems (Extended Data Fig. 5a,b), this ammonium enrichment mechanism is probably conserved among these organisms[35], contributing to the biogeochemically important nitrogen cycle (Supplementary Fig. 3). A comprehensive understanding of these biogeochemical processes is critical for preserving the vital ecological functions that sustain life on Earth.

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

## Methods

### Growth of *N. maritimus* cells

A culture of *N. maritimus* (SCM1) was provided by F. Elling and A. Pearson. Continuous cultures of SCM1 were grown on modified synthetic *crenarchaeota* medium supplemented with 1 mM $NH_4Cl$, as previously described[7,40] at 28 °C in a standing incubator that was covered to prevent excessive exposure to light. The growth of SCM1 *N. maritimus* cells was monitored using a $NO_2^-$-detection assay reported previously[1]. The same assay was used to follow SCM1 growth with differing ammonium concentrations.

### Purification of *N. maritimus* cell envelopes

Native cell envelopes were purified from *N. maritimus* by adapting a previously described protocol[23]. A total of 12 l of *N. maritimus* cultures was prepared and late-log-phase cells were collected by centrifugation (10,000*g*, 4 °C, 30 min) and frozen and stored at −80 °C until further experimentation. The cell pellet from a 1 l culture was carefully resuspended in 3 ml lysis buffer (50 mM HEPES/NaOH pH 7.5, 500 mM NaCl, 50 mM $MgCl_2$, 10 mM $CaCl_2$, 1% (w/v) CHAPS, supplemented with 1× cOmplete protease inhibitor cocktail (Roche)). The cell suspension was incubated for 1 h on ice, and then lysed using sonication (10×, 5 s pulse, amplitude strength 10%). The sonicated sample was subsequently centrifuged (80,000*g*, 4 °C, 1 h), forming a very small white pellet at the bottom of the centrifugation tube. The pellet was resuspended into 40 µl of the same buffer and used for cryo-EM experiments. For the ammonium-enriched sample, the buffer was supplemented with 2.5 mM $NH_4Cl$.

### Cryo-EM and cryo-ET sample preparation

For cryo-EM and cryo-ET grid preparation, previously reported protocols were used[18,23,41]. In brief, 2.5 µl of the specimen was applied to a freshly glow discharged Quantifoil R2/2 Cu/Rh 200 mesh grid, adsorbed for 60 s, blotted for 4–5 s and plunge-frozen into liquid ethane in the Vitrobot Mark IV (Thermo Fisher Scientific), while the blotting chamber was maintained at 100% humidity at 10 °C. For tomography, the specimen was additionally supplemented with 10 nm gold conjugated with protein A. The grids were clipped and stored under liquid nitrogen until cryo-EM data collection was performed.

### Cryo-ET and cryo-EM data collection

**Cryo-ET data.** For high-resolution in situ structure determination of the S-layer, a pipeline for high-throughput data collection was adopted[42]. In brief, a Titan Krios microscope was used to collect tilt-series data with a dose-symmetric tilting scheme[43]. Tilt series were collected at a pixel size of 1.327 Å, with a total dose of ~121 e⁻ Å⁻² applied over entire series collected between ±60° with 3° tilt increments. A total of 160 tilt series were collected with a defocus range of between −2 and −5 µm target defocus, and the samples were subjected to 0.9 s of exposure per tilt video containing 10 frames each (Extended Data Table 1). For visualization of the cellular ultrastructure, tilt-series images were acquired using the SerialEM software[44] at a pixel size 3.468 Å with a defocus range of −3 to −10 µm, ±60° oscillation, 1° increments with a total dose of ~172 e⁻ Å⁻² as well as at a pixel size of 1.33 Å with a defocus range of −4 to −10 µm, ±60° oscillation, 2° increments with a total final dose of ~160 e⁻ Å⁻².

**Cryo-EM single-particle data.** Single-particle cryo-EM data were collected as described previously[16,18,23] on the Titan Krios G3 microscope (Thermo Fisher Scientific) operating at 300 kV fitted with a Quantum energy filter (slit width 20 eV) and a K3 direct electron detector (Gatan) with a sampling pixel size of 0.546 Å running in counting super-resolution mode. For the *N. maritimus* purified sheets sample, a total of 12,557 videos over three sessions was collected with a dose rate of around 3.5 e⁻ per super-resolution pixel per s on the camera level. The sample was subjected to 4.2 s of exposure, during which a total dose of around 48–51 e⁻ Å⁻² was applied, and 40 frames were recorded per video (Extended Data Table 2).

### Subtomogram averaging of whole cells for structure determination

To obtain initial lattice maps, a previously described strategy was used[16], in which tilt-series alignment using gold fiducials and tomogram generation was performed using IMOD[45] and initial contrast transfer functions (CTFs) were estimated using CTFFIND4[46]. Tomograms for visualization were generated using the simultaneous iterative reconstruction technique (SIRT) implemented in Tomo3D[47] and denoised using Cryo-CARE[36,37]. Subtomogram averaging was performed using custom scripts written in MATLAB, described in detail elsewhere[42,48]. For initial cryo-ET structure determination, we used previously published methods[17], with the major difference being the use of a recently developed 3D-CTF correction method for tomographic data[49]. The roughly aligned subtomogram coordinates were then imported into RELION-4 for further analysis[19]. We used the tilt series after video frame alignment from the initial analysis above, without additional preprocessing, along with the tilt-series alignments performed within IMOD, CTF parameters from CTFFIND4[46] and the Euler angle assignments and subtomogram coordinates from the original analysis to proceed with refinement. The imported parameters into RELION-4 were used for multiple cycles of pseudosubtomogram generation and realignment as described recently[19]. Accounting for per-particle motions with additional cycles of pseudosubtomogram improvements and realignments increased the resolution of the *Nm*SLP hexamer to 3.4 Å in $C_6$ symmetry. Relaxation of the symmetry[50,51] led to an improved (3.3 Å) resolution overall, and 3.2 Å at the pseudohexameric axis, but decreased the resolution (~4.5 Å) at the periphery of the hexamer (Extended Data Table 1 and Extended Data Fig. 1). For spatial analysis of hexameric and pentameric S-layer positions with respect to the cell centre, the distance of each position from the cell centre was normalized by the maximally distanced hexamer/pentamer in every cell in the tomogram.

### Cryo-EM single-particle analysis

For S-layer structure from two-dimensional sheets, cryo-EM data processing was performed as described previously for S-layers in our laboratory[18,23]. Videos collected at the scope were clustered into optics groups based on the XML metadata of the data-collection software EPU (Thermo Fisher Scientific) using a *k*-means algorithm implemented in EPU_group_AFIS (https://github.com/DustinMorado/EPU_group_AFIS). Imported videos were motion-corrected, dose-weighted and Fourier cropped (2×) with MotionCor2[52] implemented in RELION-3.1[53]. CTFs of the resulting motion-corrected micrographs were estimated using CTFFIND4[46]. Initially, side views of S-layer sheets were first manually picked along the edge of the lattice using the helical picking tab in RELION while setting the helical rise to 60 Å. Top and tilted views were manually picked at the central hexameric axis. Manually picked particles were extracted in 4× downsampled 128 × 128 px² boxes and classified using reference-free 2D classification inside RELION-3.1. Class averages centred at a hexameric axis were used to automatically pick particles inside RELION-3.1. Automatically picked particles were extracted in 4× downsampled 128 × 128 px² boxes and classified using reference-free 2D classification. Particle coordinates belonging to class averages centred at the hexameric axis were used to train TOPAZ[54] in 5× downsampled micrographs using the neural network architecture conv127. For the final reconstruction, particles were picked using TOPAZ and the previously trained neural network above. Furthermore, top, bottom and side views were picked using the reference-based autopicker inside RELION-3.1, which TOPAZ did not readily identify. Particles were extracted in 4× downsampled 128 px × 128 px boxes and classified using reference-free 2D classification inside RELION-3.1. Particles belonging to class averages centred at the pseudohexameric axis were combined, and particles within 30 Å were removed to prevent

duplication after alignment. All of the resulting particles were then re-extracted in 4× downsampled $128 \times 128$ px$^2$ boxes. All of the side views and a subset of the top and bottom views were used for initial model generation in RELION-3.1. The scaled and low-pass filtered output was then used as a starting model for 3D auto refinement in a $512 \times 512$ px$^2$ box. Per-particle defocus, anisotropy magnification and higher-order aberrations[55] were refined inside RELION-3.1, followed by three rounds of focused 3D autorefinement. Bayesian particle polishing was performed subsequently in a 640 px × 640 px box[55] followed by autorefinement and symmetry relaxation[50,51]. The final map was obtained from 354,860 particles and post-processed using a soft mask focused on the central hexamer, yielding a global resolution of 2.7 Å according to the Fourier shell correlation criterion between two independently refined half-maps at a threshold value at 0.143 (ref. 56) and a local resolution of up to 2.5 Å (Extended Data Fig. 3 and Extended Data Table 2). The two-dimensional sheet-like arrangement led to anisotropy in resolution, with lower resolution perpendicular to the plane as estimated by directional FSCs[57]. Further details are provided in Extended Data Table 2 and Extended Data Fig. 3.

## Data visualization, analysis and model building
For model building, a previously described strategy was used[18,23]. For the single-particle cryo-EM map, the original $640 \times 640 \times 640$ voxel box was cropped into a $320 \times 320 \times 320$ voxel box. In both the cryo-ET and cryo-EM maps, and the protein backbone of NmSLP was manually traced as a poly-alanine model through a single NmSLP subunit using Coot[58]. Side chains were assigned at clearly identifiable positions which allowed deduction of the protein sequence register. The model was then placed into the hexameric map as six copies and subjected to several rounds of refinement using refmac5[59] inside the CCP-EM software suite[60] and PHENIX[61], followed by manually rebuilding in Coot[58]. At the N terminus, the $C_2$ maps were better resolved compared to the $C_6$ maps at the C termini of NmSLPs; therefore, multimap atomic model refinement was performed in servalcat[62]. Model validation was performed in PHENIX and CCP-EM, and data visualization was performed in UCSF Chimera[63], UCSF ChimeraX[64] and PyMOL[65]. To analyse lattice interfaces, multiple copies of the hexameric structure were placed in the cryo-EM map prepared with a larger box size. Figure panels containing cryo-EM or cryo-ET images were prepared using IMOD and Fiji[66]. Lattice maps of S-layers for visual inspection were plotted inside UCSF Chimera[63] with the PlaceObject plugin[67] and model coordinates were plotted inside UCSF ChimeraX[64] with the sym function and the BIOMATRIX PDB file header or directly using the ArtiaX plugin[68]. The SPA and STA maps were postprocessed using deepEMhancer[38] for visualization of the N-glycan densities (Fig. 2k and Extended Data Fig. 4g–h,k–l). Composite maps from focused refinements of the two-fold ($C_2$) and six-fold ($C_6$) symmetrized maps were generated using refmac5[59] and PHENIX[61] and then converted using mtz2mrc implemented in PHENIX[61].

## Bioinformatic analysis
A previously described strategy for detection and analysis of SLPs was used[18,23]. All sequence similarity searches were performed in the MPI Bioinformatics Toolkit[69] using BLAST[70] and HHpred[71]. BLAST searches were performed against the nr_arc database, a specialized subset of the NCBI non-redundant protein sequence database filtered specifically for archaeal sequences, using the default settings to identify homologues of NmSLP in archaea. The searches were seeded with the protein sequence of N. maritimus SLP. The domain organization of several obtained matches and many experimentally characterized SLPs (Supplementary Table 1) were analysed using HHpred searches with the default settings over the PDB70 and ECOD70 databases, which are versions of the PDB and ECOD databases filtered for a maximum pairwise identity of 70%, and using structural models built using Alpha-Fold (v.2.2.0)[72]. Signal peptides were predicted using SignalP (v.6.0)[73].

The mean local charge of the protein sequences was calculated using the EMBOSS charge tool[74], using a window length of 7.

## ITC analysis
ITC measurements were made using Malvern Panalytical ITC200 instruments at 25 °C in SCM buffer without ammonium chloride. Experiments were performed at a reference power of 10 µcal s$^{-1}$ and with injections at 300 s intervals to capture the large exothermic heats and broad peak profiles. The ITC cell contained N. maritimus at an optical density at 600 nm (OD$_{600}$) of 1.0 and the syringe contained 10 mM ammonium chloride in the SCM buffer. In total, ten injections, with the first injection corresponding to 0.5 µl, followed by nine injections of 1 µl were performed, resulting in a final ammonium chloride concentration of 0.475 mM in the ITC cell. N. maritimus cells were pretreated with 0, 2.5 or 5 mM EGTA for 30 min, and were then centrifuged at 16,000g for 15 min and resuspended in SCM medium lacking ammonium chloride to recover before adjusting to an OD$_{600}$ of 1.0. Control measurements of injections of ammonium chloride into buffer were performed and these heats were small and close to the values seen for buffer into buffer control experiments. This control heat was subtracted from the N. maritimus experiments before peak integration using Malvern Panalytical PEAQ software. Experiments were performed three times with different batches of N. maritimus prepared from cells in log-growth phase in SCM medium with 1 mM ammonium chloride as nutrient source. These cultures were centrifuged and resuspended in SCM buffer lacking ammonium chloride before adjusting to an OD$_{600}$ of 1 and loading into the ITC cell.

## MD simulations
The NmSLP hexamer structure was prepared for atomistic MD simulation using VMD (v.1.94)[67]. The system was first solvated with TIP3P water molecules and 0.5 M NaCl to mimic the salinity of sea water. Next, 312 ammonium ions (0.1 M NH$_4^+$) were randomly distributed throughout the solvent, along with an equal number of chloride counter ions to maintain a charge neutral system. Simulation parameters for NH$_4^+$ were derived through analogy with existing CHARMM parameters for methylammonium. Note that, to better help identify specific ion binding sites, no structural ions apparent from the NmSLP cryo-EM and cryo-ET structures were included. The resulting system contained 566,371 atoms, including 136,236 protein atoms, 141,657 water molecules, 2,246 sodium ions, 1,358 chloride ions and 312 ammonium ions, within a hexagonal box of dimensions $x = y = 217$ Å, $z = 150$ Å and axial angles $\alpha = \beta = 90°$, $\gamma = 60°$. The geometry of the simulation box was chosen so that the molecular interfaces observed between neighbouring hexamers in our tomography data would be reproduced through the interactions of the NmSLP hexamer with its periodic images in the $x–y$ plane. The system was then subjected to a series of conjugant gradient energy minimizations followed by three 500 ns MD simulations. To prevent potential distortions in the NmSLP hexamer due to the absence of structural ions offsetting its highly negative charge, protein atoms (excluding hydrogens) were harmonically restrained during simulation. Unless otherwise indicated, analyses were performed after disregarding the first 100 ns of each simulation to ensure equilibrium sampling. To assess the robustness of the observed ammonium-binding pattern, we further constructed hexamer systems containing ammonium at concentrations of 0.05 M (156 NH$_4^+$ ions) and 0.2 M (624 NH$_4^+$ ions) using an identical procedure, and these systems were subjected to a single 500 ns production simulation. Note that lower concentrations of ammonium ions could not be used due to few total ammonium ions in the box in every simulation. All simulations were conducted using NAMD (v.2.14)[68] and the CHARMM36 force field[38]. Production simulations used the NPT ensemble with conditions maintained at 1 atm and 310 K using the Nosé–Hoover Langevin piston and Langevin thermostat, respectively. The r-RESPA integrator scheme was used with an integration time step of 2 fs and SHAKE constraints applied to all hydrogen

atoms. Short-range, non-bonded interactions were calculated every 2 fs with a cut-off of 12 Å; long-range electrostatics were evaluated every 6 fs using the particle-mesh-Ewald method. Further details are provided in Supplementary Table 3.

## Reporting summary

Further information on research design is available in the Nature Portfolio Reporting Summary linked to this article.

## Data availability

Maps have been deposited at the Electron Microscopy Data Bank under accession codes EMDB-16482, EMDB-16483, EMDB-16484, EMDB-16486, EMDB-16487, EMDB-16489 and EMDB-16492. Model coordinates have been deposited at the Protein Data Bank under accession codes 8C8L, 8C8K, 8C8M, 8C8N, 8C8O and 8C8R. Further details are provided in Extended Data Tables 1 and 2. Source data are provided with this paper.

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

**Acknowledgements** This work was supported by the Medical Research Council, as part of UK Research and Innovation (programme MC_UP_1201/31 to T.A.M.B., U105184326 to J.L.). T.A.M.B. thanks the Human Frontier Science Program (grant RGY0074/2021), the Vallee Research Foundation, the European Molecular Biology Organization, the Leverhulme Trust and the Lister Institute for Preventative Medicine for support; V.A. thanks A. Lupas for continued support and the Human Frontier Science Program (grant RGY0074/2021); C.K.C. thanks P. Zhang and M. S. P. Sansom for their support as well as funding through the ERC AdG Program (grant 101021133) and a faculty start-up package from the University of Missouri-Columbia Department of Physics. We thank F. Elling and A. Pearson for the gift of a running *N. maritimus* cell culture; R. Rachel, S. H. W. Scheres and J. Zivanov for advice; and T. Darling, J. Grimmett, I. Clayson and J. J. E. Caesar for help with high-performance computing. One dataset for cryo-ET was acquired at the cryo-electron microscopy platform of the European Molecular Biology Laboratory (EMBL) in Heidelberg. This work was partly supported by institutional funds of the Max Planck Society; iNEXT, project number 653706, funded by the Horizon 2020 program of the European Union; and the MRC Laboratory of Molecular Biology Electron Microscopy Facility and Central Oxford Structural Molecular Imaging Centre (COSMIC). Simulations were performed on computational resources provided by HECBioSim, the UK High End Computing Consortium for Biomolecular Simulation, which is supported by the EPSRC (EP/L000253/1), as well as by the Research Computing Support Services division at the University of Missouri-Columbia, which is supported in part by the National Science Foundation (grant CNS-14229294). For the purpose of open access, the MRC Laboratory of Molecular Biology has applied a CC BY public copyright license to any Author Accepted Manuscript version arising.

**Author contributions** A.v.K., J.L. and T.A.M.B. designed research. A.v.K, S.v.D., L.L.P., Z.F. and T.A.M.B. performed cryo-EM and cryo-ET experiments. C.K.C. and P.J.S. performed MD simulations. A.v.K. and C.B. performed ITC measurements. V.A. performed bioinformatics analyses. A.v.K., C.K.C., S.v.D., V.A., P.J.S. and T.A.M.B. analysed data. A.v.K. and T.A.M.B. wrote the manuscript with the support of all of the authors.

**Competing interests** The authors declare no competing interests.

**Additional information**
**Correspondence and requests for materials** should be addressed to Tanmay A. M. Bharat.

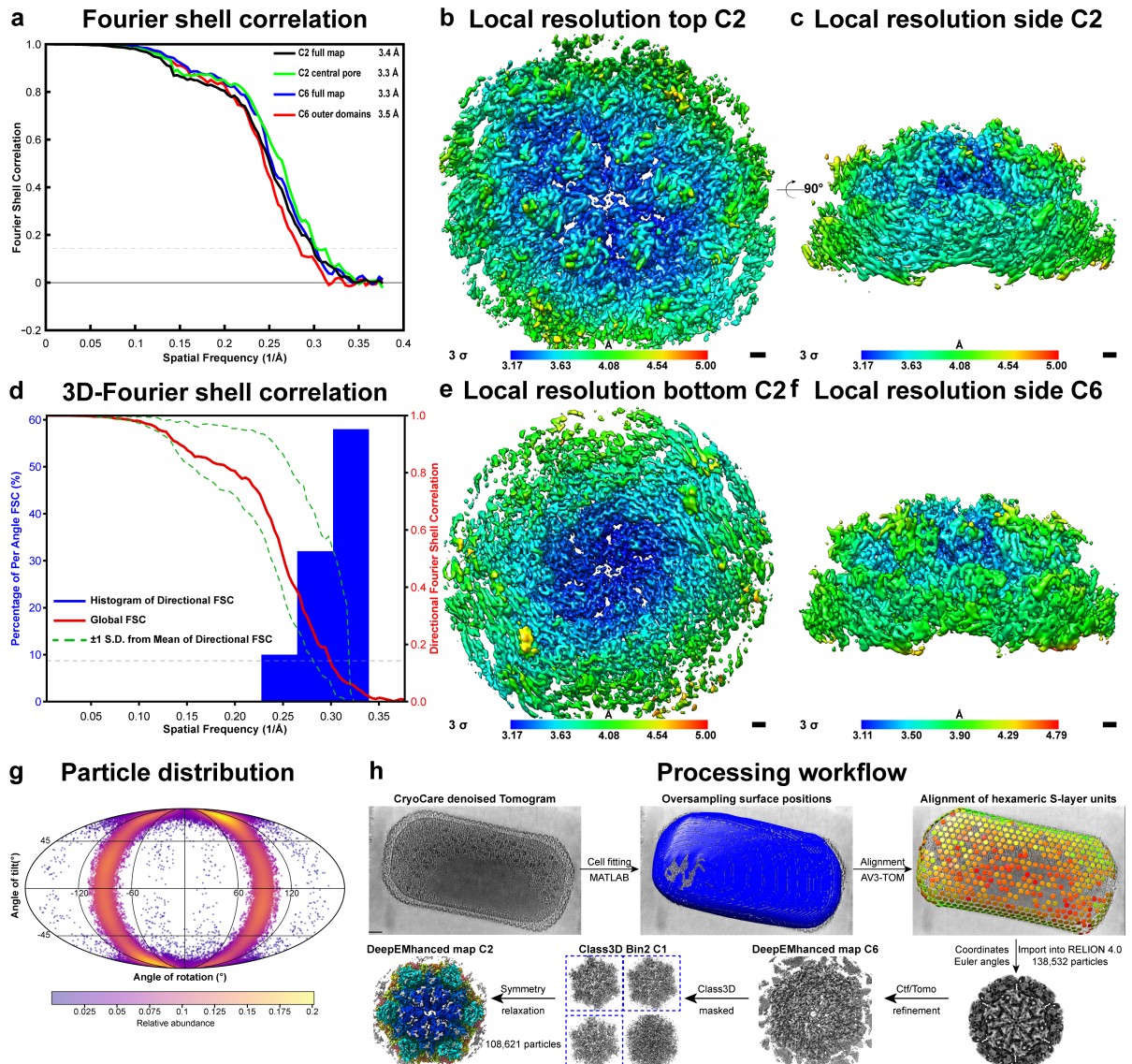

**Extended Data Fig. 1 | Subtomogram averaging (STA) of the *N. maritimus* S-layer. a**, Fourier shell correlation (FSC) curves of the STA reconstruction. **b-c**, Local resolution of two-fold symmetrised (C2) *N. maritimus* S-layer plotted onto the cryo-ET STA map, shown from the top and from the side. **d**, 3-D Fourier shell correlation (FSC) curves of the STA reconstruction. **e**, Local resolution of two-fold symmetrised (C2) *N. maritimus* S-layer shown from the bottom, from the inside of the cell. **f**, Local resolution of six-fold symmetrised (C6) *N. maritimus* S-layer map shown from the side. **g**, Particle distribution from the 0° projection image. **h**, Processing schematic from tomographic reconstructions to high resolution reconstruction of the S-layer. Scale bars for panels (**b**), (**c**), (**e**) and (**f**): 10 Å.

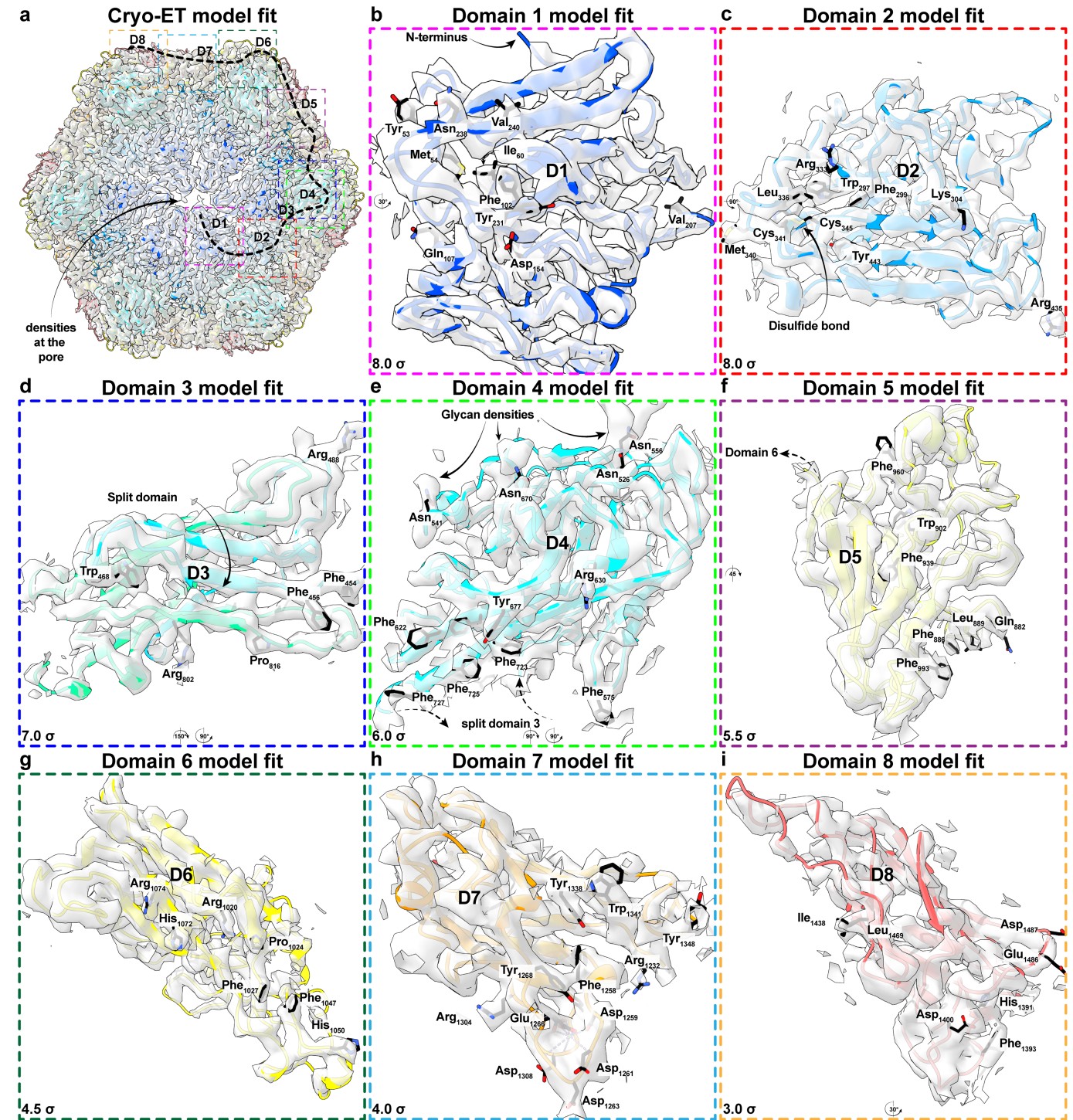

**Extended Data Fig. 2 | Structure determination from whole-cell cryo-ET data. a**, The STA map of the S-layer (isosurface shown) was used to build a model of *Nm*SLP (ribbon) directly from cellular data. (See also Fig. 1(b–e)). **b-i**, Examples of cryo-ET density and the built model for the Ig-like domains one to eight (D1-D8) of one *Nm*SLP subunit. The local resolution decreases from the central two-fold axis of the *Nm*SLP hexamer (see Extended Data Fig. 1).

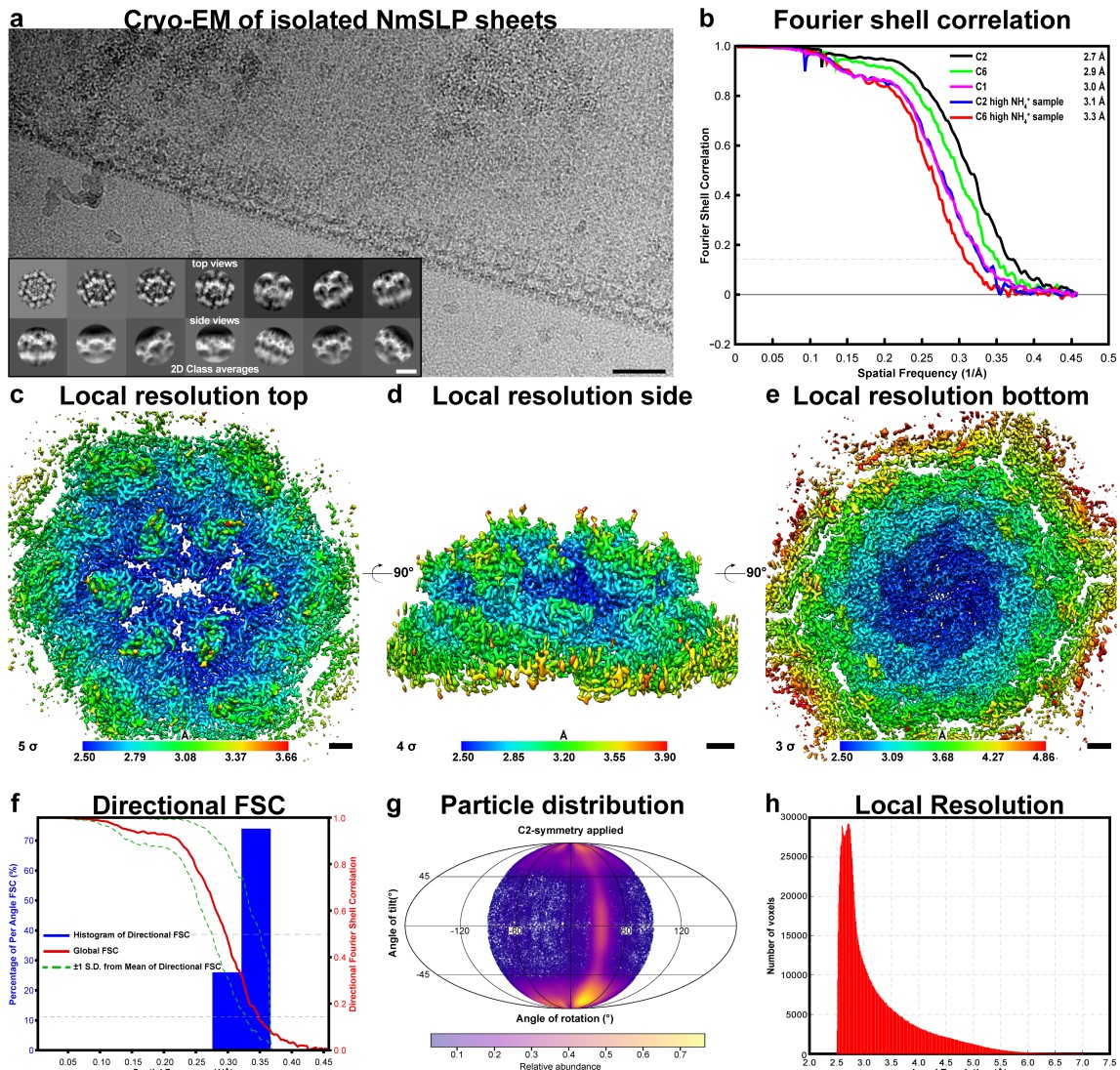

**a** Cryo-EM of isolated NmSLP sheets

**b** Fourier shell correlation

| | |
|---|---|
| C2 | 2.7 Å |
| C6 | 2.9 Å |
| C1 | 3.0 Å |
| C2 high NH4+ sample | 3.1 Å |
| C6 high NH4+ sample | 3.3 Å |

**c** Local resolution top

**d** Local resolution side

**e** Local resolution bottom

**f** Directional FSC

**g** Particle distribution

**h** Local Resolution

**Extended Data Fig. 3 | Single particle analysis of isolated *N. maritimus* S-layer sheets. a**, Cryo-EM image of isolated *N. maritimus* cell envelopes show repeating units of the pseudo-hexagonal (tilted) S-layer. Insets – characteristic top and side views observed in class averages. This single image is representative from a data set containing 12,557 images from three independent data collections (see Extended Data Table 2). **b**, FSC estimation of the resolution of the unsymmetrized (C1), two-fold (C2) and six-fold (C6) symmetrised maps. **c-e**, Local resolution of the C2 cryo-EM map estimated in RELION, plotted into the density, shown in from the top (**c**), side (**d**) and bottom (**e**). The resolution of outer domains (D8-D9) is slightly lower. **f**, Directional 3D-FSC between two random halves of the data[57]. **g**, Angular distribution of the particles in the data set, shown on a relative scale (purple denotes low and yellow denotes high). **h**, Histogram of local resolutions in voxels of the cryo-EM map (C2-symmetrised). Scale bars: (**a**) 500 Å, Inset 200 Å; (**c-d**) 20 Å.

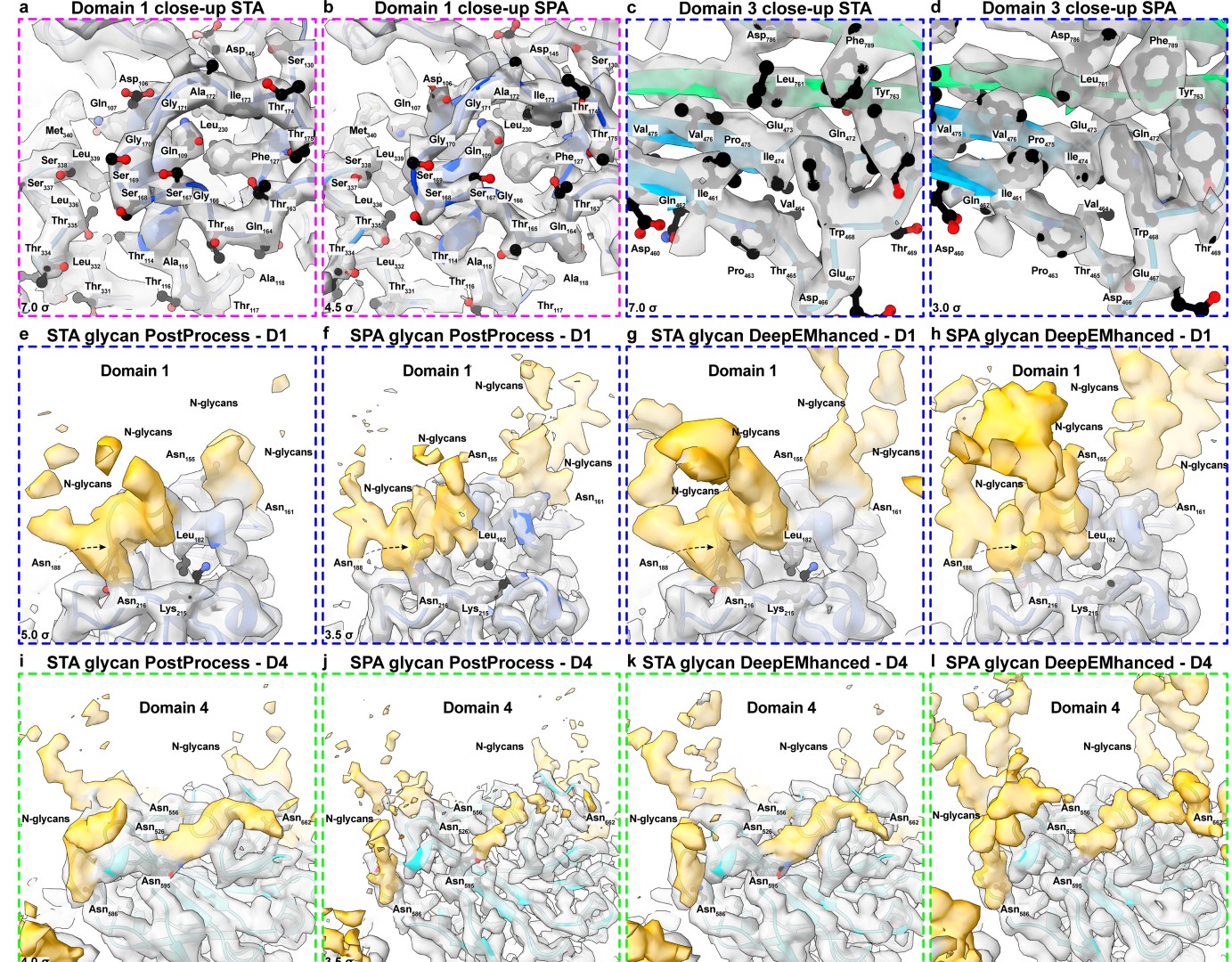

**Extended Data Fig. 4 | Comparison of the STA and SPA *N. maritimus* S-layer reconstruction. a**, Close-up view of the STA map with the built *Nm*SLP model (domain 1) showing resolved large and small side chains. **b**, Close-up view of the SPA map in the same view as shown in panel (**a**). **c-d**, Close-up view of the STA map (**c**) and (**d**) of domain 3 of the *Nm*SLP with resolved small and bulky side chains. **e-f**, Close-up views of the N-glycan densities of the STA map (**a**) and SPA map (**f**) in domain 1. **g-h**, The density of the N-glycans is enhanced in the sharpened[38] map (Methods) of the same region. **i-l**, Close-up views of some of the N-glycan densities of the STA (**i,k**) and SPA map (**j,l**) of domain 4.

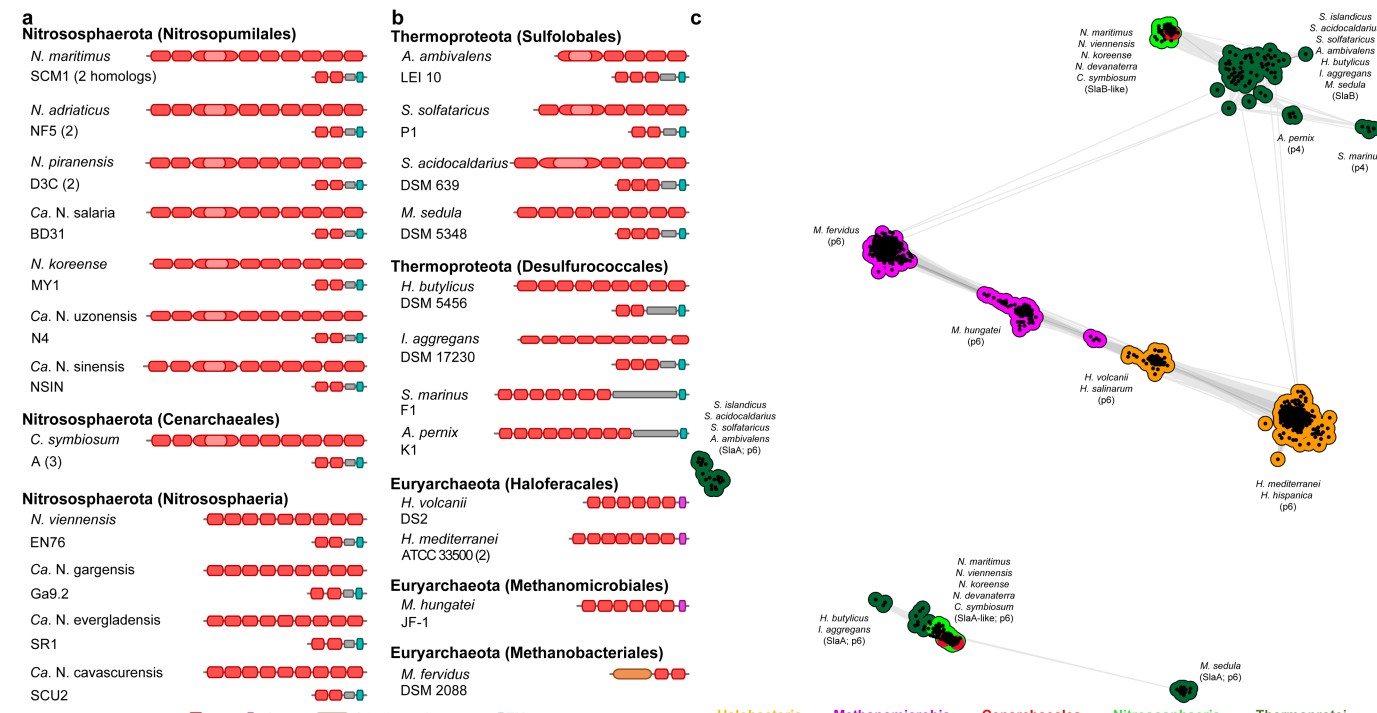

**a**
**Nitrososphaerota (Nitrosopumilales)**

*N. maritimus*
SCM1 (2 homologs)

*N. adriaticus*
NF5 (2)

*N. piranensis*
D3C (2)

*Ca.* N. salaria
BD31

*N. koreense*
MY1

*Ca.* N. uzonensis
N4

*Ca.* N. sinensis
NSIN

**Nitrososphaerota (Cenarchaeales)**

*C. symbiosum*
A (3)

**Nitrososphaerota (Nitrososphaeria)**

*N. viennensis*
EN76

*Ca.* N. gargensis
Ga9.2

*Ca.* N. evergladensis
SR1

*Ca.* N. cavascurensis
SCU2

**b**
**Thermoproteota (Sulfolobales)**

*A. ambivalens*
LEI 10

*S. solfataricus*
P1

*S. acidocaldarius*
DSM 639

*M. sedula*
DSM 5348

**Thermoproteota (Desulfurococcales)**

*H. butylicus*
DSM 5456

*I. aggregans*
DSM 17230

*S. marinus*
F1

*A. pernix*
K1

**Euryarchaeota (Haloferacales)**

*H. volcanii*
DS2

*H. mediterranei*
ATCC 33500 (2)

**Euryarchaeota (Methanomicrobiales)**

*M. hungatei*
JF-1

**Euryarchaeota (Methanobacteriales)**

*M. fervidus*
DSM 2088

Ig-like  PGF-TM  β-helix  Coiled coil  TM

**c**

Halobacteria  Methanomicrobia  Cenarchaeales  Nitrososphaeria  Thermoprotei

**Extended Data Fig. 5 | Bioinformatic analysis of Ig-like domain-containing archaeal SLPs. a**, Cartoon schematic of the domain architecture of SLPs in Nitrososphaerota. In addition to the main SLP, a minor SlaB homologue is encoded in the genome of most Nitrososphaerota. The SlaB protein exhibits a C-terminal transmembrane domain which potentially anchors the main S-layer canopy to the membrane. **b**, Cartoon schematic of the domain architecture of SLPs in Thermoproteota and Euryarchaeota. The domain architecture of AOA SLPs vary considerably from haloarchaea. **c**, Cluster map of Ig-like domain-containing archaeal SLPs. This map was created by collecting homologues of various representative Ig-like domain-containing SLPs and clustering them using CLANS[75] based on the strength of their all-against-all pairwise BLAST P-values, with a threshold set at 1-e[8]. Each protein sequence in the map is depicted as a dot, and sequences within the same taxonomic class are denoted by the same colour. The intensity of the line colour indicates the significance of sequence similarities, with darker lines representing higher significance. Although comprising Ig-like domains, archaeal SLPs are extremely divergent in their sequences.

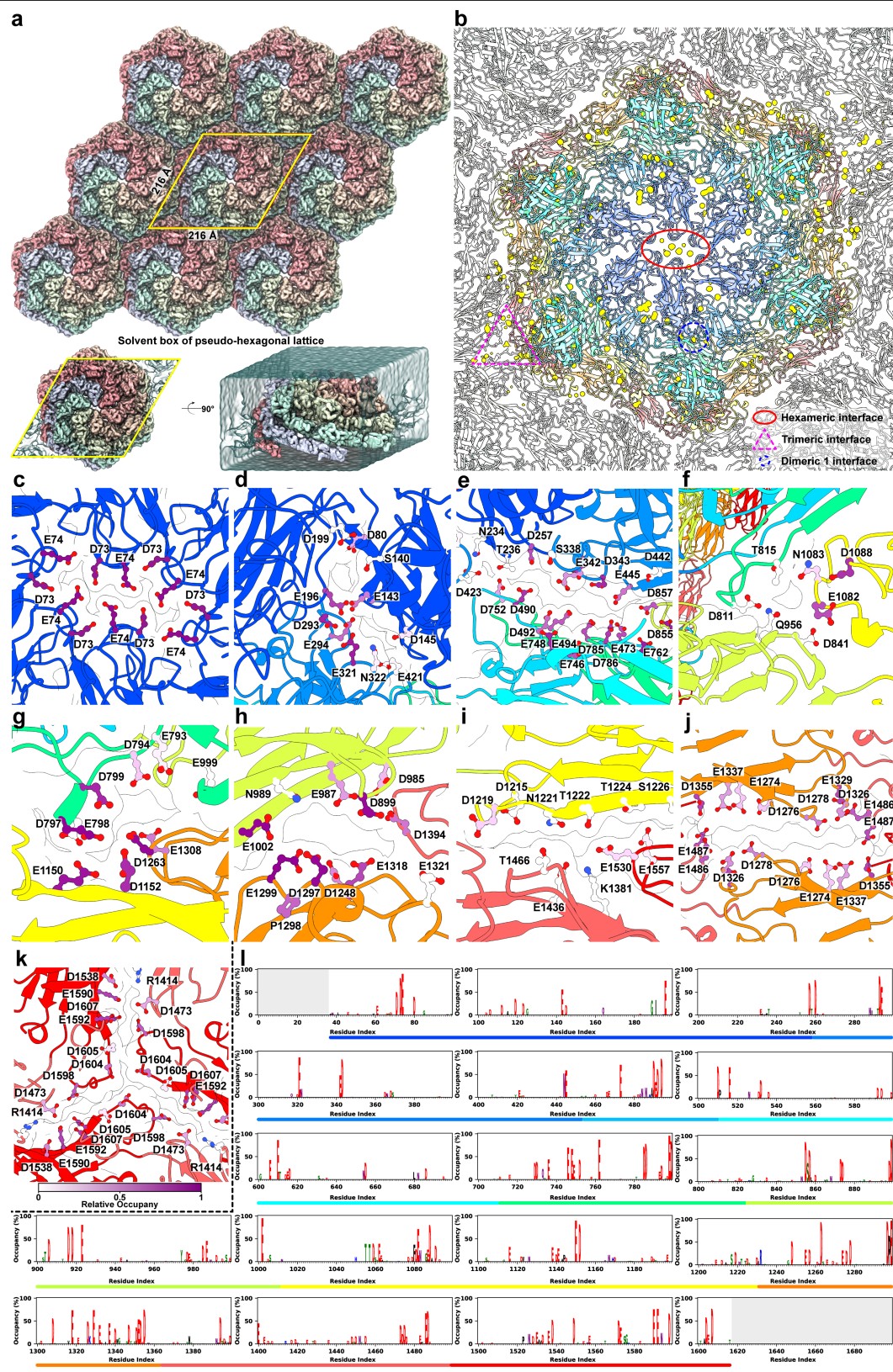

**Extended Data Fig. 6 | Molecular dynamics simulation of ammonium ion diffusion across the multi-channel, charged S-layer. a**, Unit cell design for MD simulation of the S-layer lattice. The unit cell (simulation box, outlined in yellow) was constructed to simulate an infinite two-dimensional sheet. **b**, Ammonium ion densities (ammonium occupancy during a single simulation shown as golden density) plotted onto the structure shown in ribbon representation. For further details on ammonium binding residues, please see Supplementary Table 2 and Supplementary Fig. 2. **c**-**k**, Pores in the S-layer (shown in Fig. 2), with pore residue side chains coloured by occupancy of ammonium ions (colour bar shown in panel k). **l**, Occupancy profile for the modelled *Nm*SLP sequence; residues binding to ammonium ions (high occupancy) are shown as large letters. Residue indices corresponding to domains 1 to 9 are depicted by coloured lines below the plot.

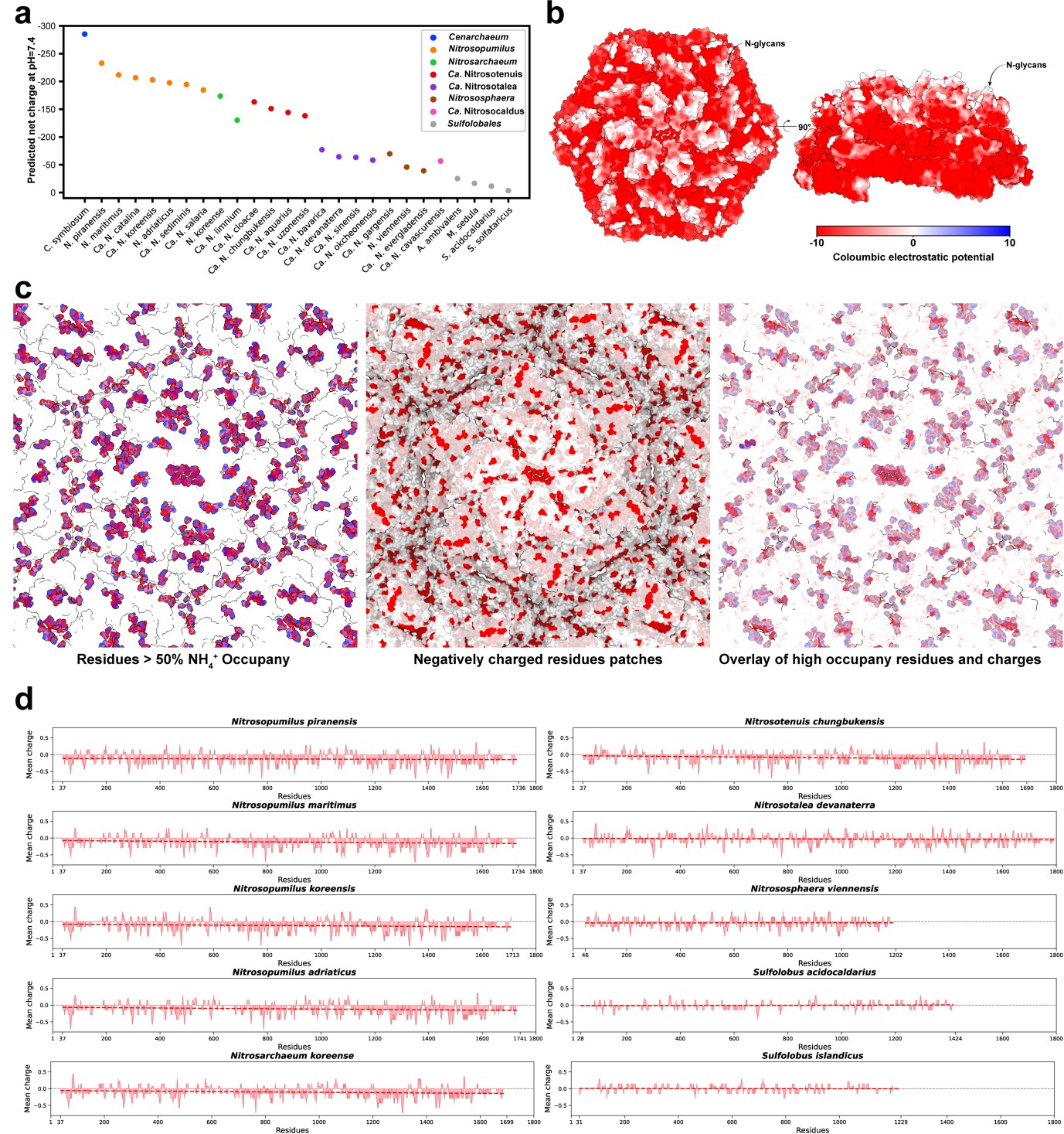

**a**

**b**

Coloumbic electrostatic potential

**c**

Residues > 50% NH₄⁺ Occupany

Negatively charged residues patches

Overlay of high occupany residues and charges

**d**

**Extended Data Fig. 7 | Charge distribution across archaeal S-layers. a**, The predicted net charges of archaeal SLPs at neutral pH (7.4) show that *N. maritimus* is highly charged, supporting its function as a cation and ammonium trap. The net charge was calculated using Isoelectric Point Calculator 2.0[76]. **b**, The electrostatic charge of the *N. maritimus* S-layer is illustrated on the surface model. **c**, Residues in the *N. maritimus* S-layer with >50% ammonium occupancy in MD simulations from pyLipID analysis (left), compared with highlighted negatively charged residues in the S-layer (middle), show a remarkable overlap (right). **d**, The mean local charge of the SLPs from AOAs and other archaea, plotted along the sequence, shows a gradual but continual increase in negative charge specifically in AOAs, but not in other archaea.

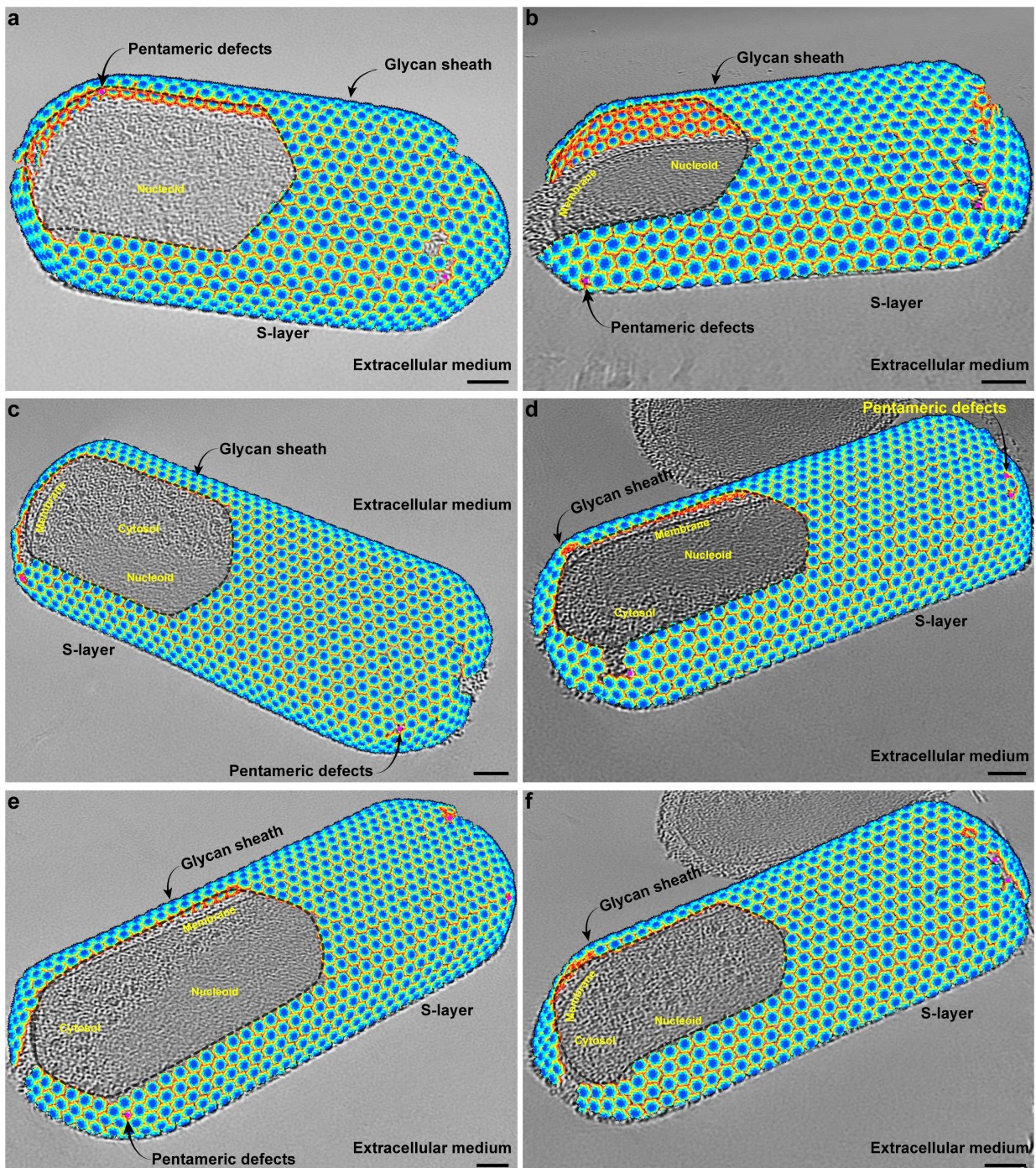

**Extended Data Fig. 8 | Whole cell tomography of *N. maritimus* and S-layer lattice maps. a-f**, Gallery of lattice maps of the S-layer from subtomogram averaging plotted onto denoised[36,37] *N. maritimus* cellular tomograms (only one slice shown in each case). The S-layer lattice coats nearly the entire outer surface of *N. maritimus* cells. A region has been cut out from each lattice map on top of the cell for clarity. The hexagonal lattice is joined together by pentameric defects, and linear lattice dislocations are also occasionally observed. Scale bars: 500 Å. Cellular tomography was performed at least 32 times (see Extended Data Table 1).

**Extended Data Table 1 | Cryo-ET data collection, refinement, and validation statistics**

| | Nmar_NmSLP_cryoET | *N. maritimus* S-layer lattice maps | *N. maritimus* S-layer lattice maps |
|---|---|---|---|
| **Data collection** | | | |
| Microscope | Titan Krios | Titan Krios | Titan Krios |
| Detector | K2 (Gatan) | K3 (Gatan) | K3 (Gatan) |
| Software | SerialEM[40] | SerialEM[40] | SerialEM[40] |
| Magnification | 105,000 | 64,000 | 26,000 |
| Voltage (kV) | 300 | 300 | 300 |
| Slit width (eV) | 20 | 20 | 20 |
| Defocus range (μm) | -2 to -5 | -2 to -10 | -3.5 to -10 |
| Pixel size (Å) | 1.327 | 1.33 | 3.468 |
| Total exposure (e$^-$/Å$^2$) | 121.36 | 159. | 170.9 |
| Exposure per tilt (e$^-$/Å$^2$) | 2.96 | 2.62 | 1.425 |
| Total number of tilts | 41 | 61 | 121 |
| Frames per tilt-movie | 10 | 10 | 3 |
| Tilt increment | ±3° | ±2° | ±1° |
| Tilt-series scheme | dose-symmetrical | dose-symmetrical | dose-symmetrical |
| Tilt range | ±60° | ±60° | ±60° |
| Tilt-series collected | 160 | 27 | 32 |
| Tilt-series used | 153 | 27 | 32 |
| **Data processing** | | | |
| Software tilt-series alignment | IMOD[41] | IMOD[41] | IMOD[41] |
| Software CTF estimation | CTFFIND4[42] | CTFFIND4[42] | CTFFIND4[42] |
| Software initial angle assignment | AV3-TOM | AV3-TOM | AV3-TOM |
| Software tilt-series refinement | RELION4.0[19] | RELION4.0[19] | RELION4.0[19] |
| Software final angle assignment | RELION4.0[19] | RELION4.0[19] | RELION4.0[19] |
| Software reconstruction | RELION4.0[19] | RELION4.0[19] | RELION4.0[19] |
| Initial particle images (no.) | 138,532 | | |
| Final particle images (no.) | 108,621 | | |
| Pre-cropped Box-size (px) | 640 x 640 x 640 | | |
| Final Box-size (px) | 200 x 200 x 200 | | |
| Pixel size final rec. (Å) | 1.327 | | |
| Symmetry imposed | C2 · C6 | | |
| Map resolution (Å) | 3.35 · 3.4 | | |
| FSC threshold | 0.143 · 0.143 | | |
| Map resolution range (Å) | 3.2-5.0 · 3.1-4.8 | | |
| Map sharpening $B$ factor (Å$^2$) | -66.19 · -72.45 | | |
| 3DFSC sphericity# | 0.94 · Not performed | | |
| EMDB code | 16487 · 16489 | | |
| Composite map EMDB | 16492 | | |
| **Model Refinement** | | | |
| Initial model used (PDB code) | None · None | | |
| Software | PHENIX[59] · PHENIX[59] | | |
| Model resolution (Å) | 3.6 · 3.9 | | |
| FSC threshold | 0.5 · 0.5 | | |
| Model composition | | | |
| Residue range | 37-445 · 446-1498 | | |
| Domain range | D1-D2 · D3-D8 | | |
| Non-hydrogen atoms | 18,347 · 47,310 | | |
| Protein residues | 2,508 · 6,318 | | |
| $B$ factors (Å$^2$) | | | |
| Protein | 74.25 · 91.26 | | |
| R.m.s. deviations | | | |
| Bond lengths (Å) | 0.002 · 0.002 | | |
| Bond angles (°) | 0.503 · 0.476 | | |
| Validation | | | |
| MolProbity score | 1.74 · 1.62 | | |
| Clashscore | 8.26 · 8.88 | | |
| Poor rotamers (%) | 0.00 · 1.14 | | |
| Cβ outliers (%) | 0.00 · 0.00 | | |
| CABLAM outliers (%) | 1.41 · 1.72 | | |
| Ramachandran plot | | | |
| Favoured (%) | 95.79 · 97.53 | | |
| Allowed (%) | 4.21 · 2.47 | | |
| Disallowed (%) | 0.00 · 0.00 | | |
| PDB code | 8C8N · 8C8O | | |
| Composite PDB code | 8C8R | | |

**Extended Data Table 2 | Cryo-EM data collection, refinement, and validation statistics**

| | NmSLP *in vitro* isolated S-layer | | | NmSLP high $NH_4^+$ | |
|---|---|---|---|---|---|
| **Data collection** | | | | | |
| Microscope | Titan Krios | Titan Krios | Titan Krios | Titan Krios | Titan Krios |
| Detector | K3 (Gatan) | K3 (Gatan) | K3 (Gatan) | K3 (Gatan) | K3 (Gatan) |
| Software | EPU | EPU | EPU | EPU | EPU |
| Magnification | 81,000 | 81,000 | 81,000 | 81,000 | 81,000 |
| Voltage (kV) | 300 | 300 | 300 | 300 | 300 |
| Electron exposure ($e^-/Å^2$) | 48.5 | 50.782 | 51.543 | 49.863 | 49.863 |
| Slit width (eV) | 20 | 20 | 20 | 20 | 20 |
| Defocus range (µm) | -1 to -3 | -1.5 to -2 | -1 to -1.75 | -0.5 to -1.75 | -0.75 to -2 |
| Acquisition Mode | Super-res | Super-res | Super-res | Super-res | Super-res |
| Pixel size (Å) | 0.546 | 0.546 | 0.546 | 0.546 | 0.546 |
| Stage tilt | 0º | 33º | 0º | 0º | 30º |
| Movies collected | 2,615 | 3,700 | 6,242 | 6,636 | 2,799 |
| Movies used | 2,575 | 3,620 | 5,455 | 6,636 | 2,799 |
| Frames per movie | 40 | 40 | 40 | 40 | 40 |
| **Data processing** | | | | | |
| Software picking | TOPAZ[52] | TOPAZ[52] | TOPAZ[52] | TOPAZ[52] | TOPAZ[52] |
| Software reconstruction | RELION3.1 | RELION3.1 | RELION3.1 | RELION3.1 | RELION3.1 |
| Initial particle images (no.) | 926,803 | 574,714 | 470,391 | 1,323,405 | 776,676 |
| Final particle images (no.) | 120,579 | 119,603 | 114,678 | 175,787 | 41,881 |
| Rescaled Box-size Class2D (px) | 128 x 128 | 128 x 128 | 128 x 128 | 128 x 128 | 128 x 128 |
| Final refinement Box-size (px) | | 640 x 640 x 640 | | 640 x 640 x 640 | |
| Pixel size final reconstruction (Å) | | 1.092 | | 1.092 | |
| Final Box-size (px) | | 640 x 640 x 640 | | 640 x 640 x 640 | |
| Final particle images (no.) | | 354,860 | | 217,668 | |
| Symmetry imposed | C1 | C2 | C6 | C2 | C6 |
| Map resolution (Å) | 3.04 | 2.71 | 2.87 | 3.05 | 3.26 |
| FSC threshold | 0.143 | 0.143 | 0.143 | 0.143 | 0.143 |
| Map resolution range (Å) | | 2.5 – 3.66 | 2.5-3.90 | 2.85-5.01 | 2.87-4.36 |
| Map sharpening *B* factor ($Å^2$) | -46.2 | -41.4 | -59.2 | -52.5 | -63.77 |
| EMDB code | | 16483 | 16482 | 16486 | |
| Composite EMDB code | | | 16484 | | |
| **Model Refinement** | | | | | |
| Initial model used (PDB code) | | None | None | | |
| Software | | Refmac5[57] | Refmac5[57] | | |
| Model resolution (Å) | | 2.9 | 3.1 | | |
| FSC threshold | | 0.5 | 0.5 | | |
| Model composition | | | | | |
| Residue range | | 37-1364 | 1365-1616 | | |
| Domain range | | D1-D7 | D8-D9 | | |
| Non-hydrogen atoms | | 59,076 | 11,430 | | |
| Protein residues | | 7,962 | 1,518 | | |
| *B* factors ($Å^2$) | | | | | |
| Protein | | 76.62 | 118.80 | | |
| R.m.s. deviations | | | | | |
| Bond lengths (Å) | | 0.014 | 0.013 | | |
| Bond angles (°) | | 1.687 | 1.619 | | |
| Validation | | | | | |
| MolProbity score | | 1.71 | 1.55 | | |
| Clashscore | | 7.99 | 10.8 | | |
| Poor rotamers (%) | | 0.76 | 0.63 | | |
| Cβ outliers (%) | | 0.33 | 0.00 | | |
| CABLAM outliers (%) | | 1.98 | 0.80 | | |
| Ramachandran plot | | | | | |
| Favoured (%) | | 95.95 | 98.07 | | |
| Allowed (%) | | 4.05 | 1.93 | | |
| Disallowed (%) | | 0.00 | 0.00 | | |
| PDB Code | | 8C8L | 8C8K | | |
| Composite PDB code | | | 8C8M | | |

# Reporting Summary

## Statistics

For all statistical analyses, confirm that the following items are present in the figure legend, table legend, main text, or Methods section.

| n/a | Confirmed | |
|---|---|---|
| ☐ | ☒ | The exact sample size (*n*) for each experimental group/condition, given as a discrete number and unit of measurement |
| ☐ | ☒ | A statement on whether measurements were taken from distinct samples or whether the same sample was measured repeatedly |
| ☐ | ☒ | The statistical test(s) used AND whether they are one- or two-sided *Only common tests should be described solely by name; describe more complex techniques in the Methods section.* |
| ☒ | ☐ | A description of all covariates tested |
| ☒ | ☐ | A description of any assumptions or corrections, such as tests of normality and adjustment for multiple comparisons |
| ☐ | ☒ | A full description of the statistical parameters including central tendency (e.g. means) or other basic estimates (e.g. regression coefficient) AND variation (e.g. standard deviation) or associated estimates of uncertainty (e.g. confidence intervals) |
| ☐ | ☒ | For null hypothesis testing, the test statistic (e.g. *F*, *t*, *r*) with confidence intervals, effect sizes, degrees of freedom and *P* value noted *Give P values as exact values whenever suitable.* |
| ☒ | ☐ | For Bayesian analysis, information on the choice of priors and Markov chain Monte Carlo settings |
| ☒ | ☐ | For hierarchical and complex designs, identification of the appropriate level for tests and full reporting of outcomes |
| ☒ | ☐ | Estimates of effect sizes (e.g. Cohen's *d*, Pearson's *r*), indicating how they were calculated |

*Our web collection on statistics for biologists contains articles on many of the points above.*

## Software and code

Policy information about availability of computer code

| Data collection | EPU v.2+, SerialEM v.3 and v.4+ |
|---|---|
| Data analysis | RELION v3.1, RELION v4.0, CTFFIND v4.1, TOPAZ v.0.2.5, IMOD v4.9.13, COOT v0.9.6, ChimeraX v1.5, Chimera v1.16, PyMOL v2.5.2, CCPEM v1.6.0, Refmac5 (Servalcat v0.2.85), PHENIX v.1.19-4092-000, VDM v.1.94,, NAMD v2.14, MATLAB v.R2019b, BLAST v.2.13.0+, HHpred v.57c87071, AlphaFold v.2.2.0, SignalP v.6.0, EMBOSS 6.6.0.0 |

For manuscripts utilizing custom algorithms or software that are central to the research but not yet described in published literature, software must be made available to editors and reviewers. We strongly encourage code deposition in a community repository (e.g. GitHub). See the Nature Portfolio guidelines for submitting code & software for further information.

## Data

Policy information about availability of data

All manuscripts must include a data availability statement. This statement should provide the following information, where applicable:

- Accession codes, unique identifiers, or web links for publicly available datasets
- A description of any restrictions on data availability
- For clinical datasets or third party data, please ensure that the statement adheres to our policy

Maps have be deposited in the Electron Microscopy Data Bank (EMDB) and atomic coordinates have been deposited in the Protein Data Bank (PDB). The accession codes are: In vitro S-layer structure (single particle analysis, SPA) with two-fold symmetry (C2): PDB ID 8C8L, EMD-16483; In vitro S-layer structure (SPA) with two-

fold symmetry (C6): PDB ID 8C8K, EMD-16482; In vitro S-layer structure (SPA) composite map: PDB ID 8C8M, EMD-16484; In vitro S-layer structure (SPA) with high NH4Cl and two-fold symmetry (C2): EMD-16486; In situ S-layer structure (subtomogram averaging, STA) with two-fold symmetry (C2): PDB ID 8C8N, EMD-16487; In situ S-layer structure (STA) with two-fold symmetry (C6): PDB ID 8C8O, EMD-16489; In situ S-layer structure (STA) composite map: PDB ID 8C8R, EMD-16492; For more details see Extended Data tables 1 and 2, respectively. No new sequences are reported in this study.

## Research involving human participants, their data, or biological material

Policy information about studies with human participants or human data. See also policy information about sex, gender (identity/presentation), and sexual orientation and race, ethnicity and racism.

| | |
|---|---|
| Reporting on sex and gender | Not applicable |
| Reporting on race, ethnicity, or other socially relevant groupings | Not applicable |
| Population characteristics | Not applicable |
| Recruitment | Not applicable |
| Ethics oversight | Not applicable |

Note that full information on the approval of the study protocol must also be provided in the manuscript.

# Field-specific reporting

Please select the one below that is the best fit for your research. If you are not sure, read the appropriate sections before making your selection.

☒ Life sciences ☐ Behavioural & social sciences ☐ Ecological, evolutionary & environmental sciences

For a reference copy of the document with all sections, see nature.com/documents/nr-reporting-summary-flat.pdf

# Life sciences study design

All studies must disclose on these points even when the disclosure is negative.

| | |
|---|---|
| Sample size | Cryo-EM and cryo-ET data set sizes were selected to obtain high-resolution reconstructions. The sample size was chosen to reach a resolution of 3.3-4.5 Å for the cryoET data and 2.7 Å for cryoEM data. |
| Data exclusions | Cryo-EM micrographs and tilt-series were selected based on high resolution content in the cryo-EM or cryo-ET workflow. Extracted particles not suitable for high-resolution reconstruction were excluded during the processing. For further details on image selection see Extended Data Tables 1 and 2. |
| Replication | The structures were solved as per the accepted protocols for data analysis, including an unbiased Fourier Shell correlation of independently aligned and averaged halves of the data. Triplicate experiments were performed for molecular simulations and growth curves, all replicates showed the same results. Other experiments like ITC were performed in triplicates. All replicates showed similar results. |
| Randomization | Not relevant to this study. Randomisation was not needed for the statistics. |
| Blinding | Not relevant to this study. Not needed for the statistics. |

# Reporting for specific materials, systems and methods

We require information from authors about some types of materials, experimental systems and methods used in many studies. Here, indicate whether each material, system or method listed is relevant to your study. If you are not sure if a list item applies to your research, read the appropriate section before selecting a response.

## Materials & experimental systems

| n/a | Involved in the study |
|-----|-----------------------|
| ☒ ☐ | Antibodies |
| ☒ ☐ | Eukaryotic cell lines |
| ☒ ☐ | Palaeontology and archaeology |
| ☒ ☐ | Animals and other organisms |
| ☒ ☐ | Clinical data |
| ☒ ☐ | Dual use research of concern |
| ☒ ☐ | Plants |

## Methods

| n/a | Involved in the study |
|-----|-----------------------|
| ☒ ☐ | ChIP-seq |
| ☒ ☐ | Flow cytometry |
| ☒ ☐ | MRI-based neuroimaging |

## Plants

| | |
|---|---|
| Seed stocks | Not relevant to this study. |
| Novel plant genotypes | Not relevant to this study. |
| Authentication | Not relevant to this study. |

