## [Peer Review File · Nature]

Manuscript Title: Membrane-less channels sieve cations in ammonia-oxidising marine archaea

Editorial Notes:

Redactions – Third Party Material

Reviewer Comments & Author Rebuttals

Reviewer Reports on the Initial Version:

Referees' comments:

Referee #1 (Remarks to the Author):

Thaumarchaeal ammonia-oxidizing archaea (AOA) are ubiquitous organisms that play a critical role in governing the global N and C cycles. Their sole energy source is ammonium. It has long been known that, in contrast to their bacterial counterpart AOB, AOA thrive in implausibly low nutrient concentrations, down to μM . Understanding the molecular mechanisms that enable AOA to tap into this scarce energy source has remained a challenge. Theoretical and numerical approaches have proposed the hypothesis that AOA evolved a cellular envelope that serves as a nutrient reservoir.

This important study goes a long way to experimentally confirming that hypothesis for *N. maritimus*. The authors solved the atomic structure of the surface layer protein (SLP) that entraps and channels ammonium ions from the oceanic environment towards the cell membrane where ammonia monooxygenase (AMO) purportedly is located. Using a high-resolution cryoET map, a single particle electron cryoEM structure, MD simulations, and ammonium-enriched cell cultures this manuscript makes the following important contributions:

1. An atomically detailed map of the S-layer of an AOA, to the best of my knowledge for the first time for any AOA
2. Resolving unambiguously the long-standing controversy about the identity of *N. maritimus* SLP, namely Nmar_1547
3. Providing experimental evidence of cation entrapment around the S-layer consistent with a previously proposed role of the S-layer as a nutrient reservoir.

Additionally, the de novo atomic structure determination technique using intact cells further refined by the authors gives direct access to the molecular mechanisms of a biogeochemically important process in situ. These and similar techniques will increasingly shed light on the molecular mechanisms of cellular processes.

This manuscript represents an original and significant scientific advance in that it provides solid experimental evidence to explain how AOA, a critical component of the global N and C cycles, can perform its function under extreme conditions. As such, the manuscript should appeal to a broad audience from molecular ecologists to structural and cellular biologists.

The conclusions are well-supported by the data presented, with potentially a minor improvement in the simulations, see below

I support publication of the manuscript with minor revisions:

P.3 line 56: ultrastructural -> I don't think this terminology is widely adopted. Superstructural?

P.5 line 97. Unexplained densities at the central C2 pore. The text refers to 'Figure S2', but I am unable to find annotated unexplained densities in the central C2 pore in Extended Data Figure 2. Is there a figure missing?

Fig 3f. What is the distance on the y-axis?

Extended Data Fig. 5: Mean local charge of NmSLP shows a gradual increase in negative charge. That is not immediately apparent from panel c. Can the authors quantify the change in charge?

P.15, line 369.

* How were the proteins prepared for MD simulations?

*I understand that 0.5M NaCl was added and then another 0.2M of NH₄⁺/Cl⁻. Was the purpose to approximate the salinity of the marine environment? Was the system appropriately neutralized by adding counterions *before* adding NaCl and NH₄⁺/Cl⁻?

Referee #2 (Remarks to the Author):

Based on mainly structural data, von Kuegelgen et al. propose a mechanism for the fundamental process of ammonium entrapment by the ammonia-oxidizing archaeon, *Nitrosopumilus maritimus*. The authors utilize a combination of cryoET and cryoEM complemented by bioinformatics and rudimentary molecular dynamics studies. The results from cryoET and cryoEM are each impressive in their own right, but the complimentary application of these techniques seems lackluster. While the whole-cell tomography provides an excellent view of the organization of the hexameric Nmar_1547 particles into the S-layer, the authors don't report any insights from their in-situ structure attained by subtomogram averaging that are not more appropriately defined in the map from in vitro reconstituted membranes attained by single particle analysis.

The resolution of the cryoET reconstruction must have been incorrectly estimated at 3.3 Å. The densities and models provided to the reviewers show no side chains except for very few bulky ones that are misleadingly highlighted in Figure 1b. This can also be seen in Extended Data Figure 2. I guess that the resolution is in the range of 4.5 Å to 5 Å instead.

Additionally, the hexagonal organization of the S-layer in *N. maritimus* appears structurally similar to previously reported archaeal S-layers including the previous 3.5 Å structure of the hexagonal S-layer lattice in *Haloferax volcanii* by the same authors. The resolution of the *N. maritimus* single particle cryo-EM reconstruction (2.7 Å) is better, however, the only major new finding is that the central channel of the hexagon is negatively charged. The authors interpret additional weak densities on the symmetry axis as cations, or ammonium ions, and try to support this assumption by MD simulations, which do not seem very professional and should be reviewed by a specialist. Until now I always thought that MD simulations describe dynamics and not binding sites. The ammonium ions

presumably follow a channel that becomes more negative toward the cell surface, and this should result in an accumulation of ammonium ions on the membrane of *N. maritimus*. As far as I know, channels cannot concentrate ions, i.e. to transport them against a gradient. In membranes this is done by transporters!

Based on these results, which in my opinion are very weak, the authors claim to have discovered how the S-layer concentrates ammonium ions on the membrane of *N. maritimus*. I do not think that their data allows this claim. Thus, this study does not provide sufficient new biological insights or technical advances over previous studies to warrant publication in a high-impact journal.

I have some comments regarding specific points:

1. A comparison of the low contour cryoEM map (Fig 2b., Extended Data Fig 3d.) with the STA map (Extended Data Fig 1c) does not show well-resolved densities for the N-glycans in the map attained by subtomogram averaging. The presence of N-glycans in the STA map at domain 4 is shown in Extended Data Fig 2e although with weaker densities than shown in the map resulting from SPA. However, in the map attained by STA as shown in supplementary video 2 many densities attributed to N-glycans are visible not only on domain 4 but also on domains 1, 2, and 5. These densities should be presented in Extended Data Fig 2. where appropriate.
2. The morph between the maps attained by SPA and STA in supplementary video 2 show that the SPA structure appears more 'relaxed' compared to the structure attained by STA. This may highlight the importance of exploring such structures in situ where physical principles such as tension are preserved in a native form. Can the authors comment on whether this or other factors contribute to the difference between the two structures?
3. Are the pentameric defects reported in Fig. 4a, Extended Data Fig. 7 observable in tomograms or only predicted geometrically? The authors have previously reported a structure of similar pentameric csg defects in the S-layer of *Haloferax volcanii*. While a complete structure of pentameric Nmar_1547 may not be necessary, some analysis of these defects on a detailed structural level would be appropriate.
4. The authors report structures of the S-layer in *N. maritimus* consisting of Nmar_1547 and not Nmar_1201. While they do explicitly state that they cannot rule out the presence of low copy number Nmar_1201, one is left wondering whether this could be confirmed through 3D classification, especially considering the high proportion of picked particles not used for the final reconstruction with SPA.
5. Line 99 in the manuscript refers to Figure S2 which purportedly shows cation densities near residues D73 and E74 in the C2 pore, but I could not find this figure. Additionally, extended Data Fig. 6d shows the residues at the pore, but the two residues mentioned in the text D73 and E74 do not appear to be in the pore region as shown here. Additionally, one would expect to also see these cation densities in Extended Data fig. 2 (or elsewhere).
6. Supplementary Video 1 shows a representative tomogram with a nicely segmented S-layer. Additionally nice would be a supplementary video of a representative tomogram without segmentation so the reader can appreciate the structure of the S-layer as seen directly in the tomogram.
7. Has the tomographic slice in Fig 1a been subject to modification eg. Denoising? The background of the tomogram in this figure differs from that shown in Supplementary Video 1. If so, this should be

reported in the Methods where appropriate.

8. The methods section for subtomogram averaging mentions the usage of custom scripts. The authors should include what these scripts were used for, and whether they are publicly available.

9. The method by which particle positions for subtomogram averaging were attained is not mentioned in the methods (unless it is part of the custom scripts referred to in comment 8).

Referee #3 (Remarks to the Author):

Kügelgen et al. present a comprehensive approach in cryo-EM, cryo-ET, bioinformatic analysis, simulations and growth experiments to characterize membrane-less channels from the ammonia-oxidizing marine archaea *Nitrosopumilus maritimus*. In their study, the authors reveal the architectures of a molecular machinery that captures and directs ammonium ions to the cell membrane. The subtomogram with a resolution of 3.3 Å is very impressive, and the 2.7 Å resolution of the single particle electron cryomicroscopy reconstruction shows in detail how the S-layer is arranged to trap ammonia ions. In summary, the authors' analysis defines the power of tomography as a technology of the future, paving a new path in structural biology. The manuscript is well written. However, some terms are used incorrectly. References are sufficient, but SI figures need improvement to support the current manuscript. I recommend publishing the manuscript in Nature if my concerns can be addressed.

1) Line 25. Many others and I always have trouble defining an atomic structure with a resolution of about 3 Å. At atomic resolution, the atoms are separated from each other, or at least water molecules or the rings of the cyclic side chains can be seen. I would change that term to "high resolution", which is a more general term and covers this range of resolutions.

2) Line 26. In my opinion, de novo always describes a new fold of a protein and it's clearly not an entirely new type of structure but an interesting novelty. Especially since there is a solved structure (pdb id 7o23) with low id compared to Nmar_1547 showing similar folding in some areas. In addition, it has been shown in many examples that S-layers are assembled in pseudo-hexagonal arrays. Therefore, the authors should elaborate the novel properties of this complex in order to bring a better understanding to the reader.

3) Line 30. Fix statement: SPA cryo-EM does not provide arrangement information of the S-layer in a lattice, but cryo-ET does.

4) Line 42. 1028 cells sound like a lot, but the number is difficult for the reader to catch. Is there another order of magnitude that gives a better idea? Maybe how many tons of Amomium are converted or what proportion of these microorganisms have compared to others? As far as is known, sequencing shows that *Nitrosopumilus maritimus* make up about 1% of the metagenome on the surface of the oceans (Walker et al. PNAS 2010).

5) Extended Data Fig. 1a&b. The FSC curve clearly shows that the STA map has a resolution of 3.3 Å. In contrast, the Resmap has the best local resolution at around 3.2 Å. That concludes for me the

overall resolution here is 0.5 Angstrom higher. And why does the scale bar range from 3.2 to 5 Å?
Please clarify.

6) Line 70. A paper - focusing on structural studies should elaborate on "computational algorithms" and thus point towards a processing scheme at this point - this is clearly missing in the manuscript

7) Line 75: pseudo-hexagonal - please add deviation from ideal C6 point group; was symmetry applied during processing?

8) Line 80. Local map quality in Extended Data Fig. 2. can hardly be assessed by the areas selected - smaller areas showing all side chains would be preferable. In addition, the authors present some side chains in the STA maps, so it would be nice to show the same for the SPA map.

9) Line 84. The map shows a preferred orientation with local anisotropy, which explains the peripheral drop in resolution, which is also reflected in the 3D Fourier shell correlation histogram. I suspect there is a lack of side views of the particles. Therefore, would it be possible to represent the angular distribution of the particles for the STA map as for the SPA map?

10) Line 94. Unexplained densities are not mentioned in Extended Data Fig. 2 - point towards the areas of interest.

11) Line 101. Include region in Extended Data Fig. 2; add a sentence of caution: at the reported local resolution of about 3.2 Å, ions are barely visible. So, I would be very careful showing that in Fig 3a.

12) Line 112. FSC curves with different point group symmetry applied are indicated, C1 reconstruction is missing. Please add.

13) Line 113. Proof of "excellent agreement" by model-superposition. RMSD of both models required and overlay of maps - side-by-side comparison of different figures is not enough.

14) Line 118. Please extend your observation of missing residues by a possible explanation, as to why this area was not resolved; might not even be a methodical limitation?

15) Line 139. What kind of glycans are present? Density in SPA data is of sufficient quality to build initial glycan trees, at least in Fig. 2b.

16) Line 150. Growth curves should be shown in color to better distinguish them.

17) Line 160 & 677. Range of APBS result for surface electrostatic potential not suitable for good representation of smaller changes in charge distribution - would be better suited to show the solvent-accessible region within the "the membrane proximal C-terminus highly negatively charged" - charge over residues plot not really fitting to what you want to communicate with your data.

18) Line 178. Please add a reference to published data or your own data – missing.

Discussion:

19) Add conclusion about importance of glycosylation - clearly not on all domains, explain inhomogeneous distribution towards the center of the structure.

20) Add a theorem about the cone-like structure of each β -sheet hexamer and the implications of such a geometry to allow dense formation of the sheet around a curved surface (the cell).

Author Rebuttals to Initial Comments:

Response to Reviewers' Comments for von Kügelgen *et al*

Black text – original reviewers' comments

Blue text – our response

Red text – manuscript excerpt

We thank the reviewers for their valuable comments, which have helped us improve the manuscript, constituting a major revision of the work.

Referee #1 (Remarks to the Author):

Thaumarchaeal ammonia-oxidizing archaea (AOA) are ubiquitous organisms that play a critical role in governing the global N and C cycles. Their sole energy source is ammonium. It has long been known that, in contrast to their bacterial counterpart AOB, AOA thrive in implausibly low nutrient concentrations, down to μM . Understanding the molecular mechanisms that enable AOA to tap into this scarce energy source has remained a challenge. Theoretical and numerical approaches have proposed the hypothesis that AOA evolved a cellular envelope that serves as a nutrient reservoir.

This important study goes a long way to experimentally confirming that hypothesis for *N. maritimus*. The authors solved the atomic structure of the surface layer protein (SLP) that entraps and channels ammonium ions from the oceanic environment towards the cell membrane where ammonia monooxygenase (AMO) purportedly is located. Using a high-resolution cryoET map, a single particle electron cryoEM structure, MD simulations, and ammonium-enriched cell cultures this manuscript makes the following important contributions:

1. An atomically detailed map of the S-layer of an AOA, to the best of my knowledge for the first time for any AOA
2. Resolving unambiguously the long-standing controversy about the identity of *N. maritimus* SLP, namely Nmar_1547
3. Providing experimental evidence of cation entrapment around the S-layer consistent with a previously proposed role of the S-layer as a nutrient reservoir.

Additionally, the de novo atomic structure determination technique using intact cells further refined by the authors gives direct access to the molecular mechanisms of a biogeochemically important process in situ. These and similar techniques will increasingly shed light on the molecular mechanisms of cellular processes.

This manuscript represents an original and significant scientific advance in that it provides solid experimental evidence to explain how AOA, a critical component of the global N and C cycles, can perform its function under extreme conditions. As such, the manuscript should appeal to a broad audience from molecular ecologists to structural and cellular biologists.

The conclusions are well-supported by the data presented, with potentially a minor improvement in the simulations, see below

Thank you for the encouraging comments.

I support publication of the manuscript with minor revisions:

P.3 line 56: ultrastructural -> I don't think this terminology is widely adopted. Superstructural?

We have amended this sentence as suggested.

“At the overall morphological scale, the *N. maritimus* S-layer..”

P.5 line 97. Unexplained densities at the central C2 pore. The text refers to 'Figure S2', but I am unable to find annotated unexplained densities in the central C2 pore in Extended Data Figure 2. Is there a figure missing?

We apologise for the confusion; we have now marked these densities in the Extended Data Fig. 2.

However, given the comments from reviewers 2 and 3, we have practiced caution in interpreting such putative ionic densities without orthogonal evidence. Which is why we have sought to provide orthogonal evidence for ammonium binding in the new Fig. 3 and Extended Data Fig. 6.

“Ammonium binding to *N. maritimus* cells, biological repeats are shown at different EGTA concentrations, which show different levels of S-layer disruption.”

Fig 3f. What is the distance on the y-axis?

To avoid misinterpretation, now we have changed this to distance from the lowest atom built in the S-layer structure, now clarified in the revised figure legend.

“A histogram of negatively charged residues along the S-layer, overlaid on the ammonium ion positions in the 0.1 M NH_4^+ MD simulations (distance is calculated from the closest, membrane proximal amino acid residue in the S-layer structure).”

Extended Data Fig. 5: Mean local charge of NmSLP shows a gradual increase in negative charge. That is not immediately apparent from panel c. Can the authors quantify the change in charge?

To alleviate this concern, we now provide a trend line in the figure panel (now Fig. 4d) to show the overall decrease in the charge.

“d, Mean local charge of NmSLP plotted along the sequence shows gradual but continual increase in negative charge (see also Extended Data Fig. 8). The Z position of the NmSLP residues, derived from the S-layer structure, is indicated, with the ninth Ig-like domain forming the base of the S-layer, proximal to the cell membrane.”

We further show similar charge distributions for other AOA and non-AOAs (in the new Extended Data Fig. 8).

“f, The mean local charge of the SLPs from AOA and other archaea, plotted along the sequence, shows a gradual but continual increase in negative charge specifically in AOA.”

These plots show that only in the case of AOA's do the negative charges increase along the sequence, whereas for non-AOA's such as *S. islandicus*, no such negative trend is apparent.

P.15, line 369.

* How were the proteins prepared for MD simulations?

Although this information was reported in the methods section, we realise that it may not have been complete. We have updated the methods section with further details. Relevant text related to protein and S-layer preparation for simulations is appended below.

“The NmSLP hexamer structure was prepared for atomistic MD simulation using VMD v1.94⁶⁵. The system was first solvated with TIP3P water molecules and 0.5 M NaCl to mimic the salinity of sea water. Afterwards, ammonium (NH_4^+) ions were randomly distributed throughout the solvent at three concentrations, 0.05 M (156 NH_4^+), 0.1 M (312 NH_4^+) and 0.2 M (624 NH_4^+), along with an equal number of chloride counter ions to maintain a charge neutral system. Simulation parameters for NH_4^+ were derived via analogy with existing CHARMM parameters for methylammonium. Note, to better help identify specific ion binding sites, no structural ions apparent from the NmSLP cryo-EM and cryo-ET structures were included. The resulting systems each contained roughly 565,000 atoms within a hexagonal box of dimensions $x = y = 217 \text{ \AA}$, $z = 150 \text{ \AA}$ and axial angles $\alpha = \beta = 90^\circ$, $\gamma = 60^\circ$. The geometry of the simulation box was chosen so that the molecular interfaces observed between neighbouring hexamers in our tomography data would be reproduced through the interactions of the NmSLP hexamer with its periodic images in the x-y plane. Each S-layer system was then subjected to a series of conjugate gradient energy minimisations followed by a 500-ns MD simulation. To prevent potential distortions in NmSLP due to the absence of structural ions offsetting its highly negative charge, protein atoms (excluding hydrogens) were harmonically restrained throughout the simulation.”

*I understand that 0.5M NaCl was added and then another 0.2M of $\text{NH}_4^+/\text{Cl}^-$. Was the purpose to approximate the salinity of the marine environment? Was the system appropriately neutralized by adding counterions *before* adding NaCl and $\text{NH}_4^+/\text{Cl}^-$?

Yes, indeed that is the case. We wanted to approximate the marine environment with the NaCl and the system was neutralised with chloride ions. We would also like to highlight that now we have performed simulations at differing ammonium concentrations to validate our results.

“The system was first solvated with TIP3P water molecules and 0.5 M NaCl to mimic the salinity of sea water. Afterwards, ammonium (NH_4^+) ions were randomly distributed throughout the solvent at three concentrations, 0.05 M (156 NH_4^+), 0.1 M (312 NH_4^+) and 0.2 M (624 NH_4^+), along with an equal number of chloride counter ions to maintain a charge neutral system.”

Referee #2 (Remarks to the Author):

Based on mainly structural data, von Kuegelgen et al. propose a mechanism for the fundamental process of ammonium entrapment by the ammonia-oxidizing archaeon, *Nitrosopumilus maritimus*. The authors utilize a combination of cryoET and cryoEM complemented by bioinformatics and rudimentary molecular dynamics studies. The results from cryoET and cryoEM are each impressive in their own right, but the complimentary application of these techniques seems lackluster. While the whole-cell tomography provides an excellent view of the organization of the hexameric Nmar_1547 particles into the S-layer, the authors don't report any insights from their in-situ structure attained by subtomogram averaging that are not more appropriately defined in the map from in vitro reconstituted membranes attained by single particle analysis.

We agree with the reviewer that more analyses using the cryoET could have been performed. To rectify this situation, we have used the whole-cell subtomogram mapping to locate pentamers in the S-layer. We found that pentamers are almost exclusively found on the cell poles, evidenced by our quantification shown below on all tomograms in the data set (presented in the new Fig. 1).

“b, Map of the subtomogram positions in the cellular S-layer, at the edge of the cell shows presence of pentameric defects. Scale bar: 500 Å. c, A histogram of subtomogram positions from all tomograms relative to the three-dimensional centre of the cell body (grey: hexamers, pink: pentamers).”

Furthermore, in the new Fig. 3, we have used cryo-ET to study the effect of S-layer disruption on the cell shape.

“d, Cryo-ET of *N. maritimus* cells after titration with ammonium chloride shows complete coating with an S-layer. e, Cryo-ET of *N. maritimus* cells treated with 2.5 mM EGTA after titration with ammonium chloride shows gaps in the S-layer in a heterogeneous cell population. f, Cryo-ET of *N. maritimus* cells treated with 5 mM EGTA after titration with ammonium chloride shows round cells with a naked membrane. Scale bars: 500 Å.”

The resolution of the cryoET reconstruction must have been incorrectly estimated at 3.3 Å. The densities and models provided to the reviewers show no side chains except for very few bulky ones that are misleadingly highlighted in Figure 1b. This can also be seen in Extended Data Figure 2. I guess that the resolution is in the range of 4.5 Å to 5 Å instead.

Sorry about the confusion. The reviewer is correct that at the edges of the map, the resolution is lower, closer to ~ 4.5 Å. However, at the centre of the hexamer the resolution is 3.3 Å, and this is all summarised in the local resolution plot shown in the Extended Data Fig. 1.

We are keen to prevent further confusion, therefore we have now explicitly mentioned this resolution anisotropy and reported the resolution range in the manuscript.

“...using electron cryotomography and subtomogram averaging, at a resolution of up to 3.3 Å from whole cells.”

and

“The central region of the S-layer hexamer in the map had a resolution of 3.3 Å, with the resolution decaying to ~ 4.5 Å towards the periphery of the hexamer (Extended Data Fig. 1).”

and

“The first eight domains were well resolved in our 3.3-4.5 Å-resolution subtomogram averaging map.”

In addition to the above clarification, we also provide a new Extended Data Figure 4 where smaller side chains are visible in the STA and SPA maps.

“Extended Data Fig. 4| Comparison of the STA and SPA *N. maritimus* S-layer reconstruction.

a, Close-up view of the STA map with the built NmSLP model (domain 1) showing resolved large and small side chains. **b**, Close-up view of the SPA map in the same view as shown in panel (a).”

Additionally, the hexagonal organization of the S-layer in *N. maritimus* appears structurally similar to previously reported archaeal S-layers including the previous 3.5 Å structure of the hexagonal S-layer lattice in *Haloferax volcanii* by the same authors.

While there are some structural similarities, the organisation of the *N. maritimus* S-layer is different from that of *Haloferax volcanii*. Additionally, while the S-layer proteins of *N. maritimus*, *H. volcanii*, and many other archaea comprise Ig-like domains, their sequences are highly divergent, and the sequence similarity between them is barely detectable, as highlighted by the new bioinformatic cluster map presented in Extended Data Fig. 5.

“c, Cluster map of Ig-like domain-containing archaeal SLPs. This map was created by collecting homologs of various representative Ig-like domain-containing SLPs and clustering them using CLANS⁷⁵ based on the strength of their all-against-all pairwise BLAST P-values, with a threshold set at 1-e8. Each protein sequence in the map is depicted as a dot, and sequences within the same taxonomic class are denoted by the same colour. The intensity of the line colour indicates the significance of sequence similarities, with darker lines representing higher significance. Although comprising Ig-like domains, archaeal SLPs are extremely divergent in their sequences.”

To summarise, S-layers from AOA (including *N. maritimus*) form a distinct cluster, set apart in sequence space from halophilic archaea. These differences manifest in various ways, including marked

changes in the lengths of SLP, an increased number of Ig-like domains in AOA, distinct cell anchoring mechanisms, varied glycosylation patterns, and the presence of a heavily glycosylated domain sandwiched within one of the Ig domains (domain 4 in *N. maritimus*).

The resolution of the *N. maritimus* single particle cryo-EM reconstruction (2.7 Å) is better, however, the only major new finding is that the central channel of the hexagon is negatively charged.

We are very sorry that this part of the manuscript was not well presented. Rather than only a single channel/pore present in the centre of the hexamer, there are pores found in all parts of the S-layer (shown in the new Fig. 2). All of these pores are all predominantly lined by negatively charged residues, supporting ammonium binding or ammonium movement through the S-layer. These are now highlighted in the new Fig. 2.

“Fig. 2| Cryo-EM structure of isolated *N. maritimus* S-layer sheets.

a, *In vitro*, 2.7 Å-global-resolution cryo-EM structure of isolated S-layer sheets from *N. maritimus*. The colour scheme for ribbon diagrams is the same as in Fig. 1, domains of one NmSLP are marked. **b-j**, Magnified views of the pores lined with negatively charged residues, which are ubiquitous in the S-layer sheet. The location of the pores is given in the titles of the panels. **k**, The sharpened⁶⁷ cryo-EM map shows seventeen glycans decorating each NmSLP, map shown at a lower contour level in two different orientations. A schematic of the glycan locations on the NmSLP sequence is shown below.”

The authors interpret additional weak densities on the symmetry axis as cations, or ammonium ions, and try to support this assumption by MD simulations, which do not seem very professional and should be reviewed by a specialist.

We agree with the reviewer that interpretation of densities at the symmetry axes could be problematic, a caution also raised by reviewer 3. To alleviate this important concern, we now provide additional, orthogonal evidence for ammonium binding by *N. maritimus* cells, which is compromised when the S-layer is disrupted (shown in the new Fig. 3). Our data show that either ammonium ions directly bind to the S-layer or pass through its pores to bind to the cell below.

“Fig. 3] Ammonium ions either bind directly to the S-layer or pass through it to bind to the underlying cell.

a, ITC signal of intact *N. maritimus* cells titrated with ammonium chloride with different pre-treatments with EGTA. **b**, Quantification of the ITC curves showing total heat released in the different experiments. Statistically significant differences are denoted with *. **c**, Schematic of the ITC and cryo-ET experiment presented. **d**, Cryo-ET of *N. maritimus* cells after titration with ammonium chloride shows complete coating with an S-layer. **e**, Cryo-ET of *N. maritimus* cells treated with 2.5 mM EGTA after titration with ammonium chloride shows gaps in the S-layer in a heterogeneous cell population. **f**, Cryo-ET of *N. maritimus* cells treated with 5 mM EGTA after titration with ammonium chloride shows round cells with a naked membrane. Scale bars: 500 Å.”

We were surprised by the reviewer's concerns regarding our MD simulations, given our team's extensive background in this area, with co-authors of this manuscript having published over 100 papers on MD simulations of membrane proteins. We assure the reviewer that methods employed in our study are indeed state-of-the-art. However, we respect and value the feedback provided. To address these concerns and reaffirm the robustness of our methods, we have nevertheless performed further simulations at different ammonium concentrations. These additional simulations consistently demonstrate increased residence times of ammonium ions at negatively charged pores in the S-layer (see new Extended Data Fig. 7).

“Extended Data Fig. 7 | Molecular dynamics simulation of ammonium ion diffusion across the multi-channel, charged S-layer.

a, Unit cell design for MD simulation of the S-layer lattice. The unit cell (simulation box, outlined in yellow) was constructed to simulate an infinite two-dimensional sheet. **b**, Ammonium ion densities (cation residence times during the simulation, golden density) plotted onto the structure shown in ribbon representation. **c-k**, Pores in the S-layer (shown in Fig. 2), with pore residue side chains coloured by

occupancy of ammonium ions (colour bar shown in panel k). I, Occupancy profile for the modelled NmSLP sequence; residues binding to ammonium ions (high occupancy) are shown as large letters.”

Until now I always thought that MD simulations describe dynamics and not binding sites.

While it is true that MD simulations provide dynamic information, residence times of ligands can be used to estimate binding. This is a standard practice in the field, some references are listed below - <https://doi.org/10.1021/acs.jctc.1c00708>
<https://doi.org/10.1038/nature12595>
<https://doi.org/10.1038/s41467-020-17437-5>
<https://www.nature.com/articles/s42004-022-00721-4>

Using residence times as a readout for binding, we can predict which residues bind to ammonium ions. Our prediction matches well with the observed negative charge distribution of the S-layer (shown in the new Fig. 4).

“Fig. 4| Ammonium (NH₄⁺) binding to the negatively charged S-layer lattice.

a, MD simulations support ammonium ion binding at the S-layer pores. Residue-based ammonium occupancies during the 0.1 M NH₄⁺ MD simulations are plotted onto the S-layer structure on a relative scale from white to purple. **b**, Distribution of negatively charged residues (shown in red) in the S-layer matches well with the MD simulations showing predicted ammonium binding residues (see also Extended Data Fig. 7). **c**, A histogram of negatively charged residues along the S-layer, overlaid on the ammonium ion positions in the 0.1 M NH₄⁺ MD simulations (distance is calculated from the closest, membrane proximal amino acid residue in the S-layer structure).”

The ammonium ions presumably follow a channel that becomes more negative toward the cell surface, and this should result in an accumulation of ammonium ions on the membrane of *N. maritimus*. As far as I know, channels cannot concentrate ions, i.e. to transport them against a gradient. In membranes this is done by transporters!

The S-layer provides multiple pores or channels for ammonium movement, rather than a single pore in the centre of the hexamer. Sorry, this was not made clear in the previous version of the manuscript. We have now shown all the pores in the revised Fig. 2, all predominantly lined by negatively charged residues, allowing ammonium entrapment.

[REDACTED]

The concentration effect is made possible by the presence of an ammonium sink in the cell membrane, which is the ammonium oxidation machinery. The presence of this sink leads to ammonium enrichment in the cell, predicted previously by theoretical bulk simulations by Li *et al*, *J. Phys. Chem. B.* (2019), see adjoining figure from that manuscript [REDACTED]

Based on these results, which in my opinion are very weak, the authors claim to have discovered how the S-layer concentrates ammonium ions on the membrane of *N. maritimus*. I do not think that their data allows this claim. Thus, this study does not provide sufficient new biological insights or technical advances over previous studies to warrant publication in a high-impact journal.

We appreciate the reviewer's insightful critique, which has prompted us to strengthen the evidence supporting our conclusions. In response, we have incorporated additional data and analyses to support the mechanism reported in this study, including new biochemical data (Fig. 3 and Extended Data Fig. 6), new MD simulations (Fig. 4 and Extended Data Fig. 7), new cryo-ET analysis related to pentamer positions (Fig. 1), and related to the effect of S-layer disruption on cell shape (Fig. 3), and new bioinformatic analysis (Fig. 4 and Extended Data Fig. 5). Our revised manuscript shows that this mechanism is probably shared by all AOA, likely playing a significant role in driving the global nitrogen cycle by ammonium trapping. Additionally, we acknowledge that our original manuscript did not sufficiently emphasize certain critical aspects, such as the ubiquitous presence of pores throughout the S-layer and the negatively charged residues lining them. These points, along with other concerns raised by the reviewers, have been thoroughly addressed in our revised manuscript, ensuring that our revised version comprehensively presents our findings and their significance.

I have some comments regarding specific points:

1. A comparison of the low contour cryoEM map (Fig 2b., Extended Data Fig 3d.) with the STA map (Extended Data Fig 1c) does not show well-resolved densities for the N-glycans in the map attained by subtomogram averaging. The presence of N-glycans in the STA map at domain 4 is shown in Extended Data Fig 2e although with weaker densities than shown in the map resulting from SPA. However, in the map attained by STA as shown in supplementary video 2 many densities attributed to N-glycans are visible not only on domain 4 but also on domains 1, 2, and 5. These densities should be presented in Extended Data Fig 2. where appropriate.

We have now highlighted these densities as requested in the new Extended Data Figure 4.

2. The morph between the maps attained by SPA and STA in supplementary video 2 show that the SPA structure appears more 'relaxed' compared to the structure attained by STA. This may highlight the importance of exploring such structures in situ where physical principles such as tension are

preserved in a native form. Can the authors comment on whether this or other factors contribute to the difference between the two structures?

In our opinion, the reviewer's inference about the S-layer curvature and tension is correct. We believe that this is indeed the major factor explaining the difference. We have now commented on this in the text.

"The NmSLP hexamer in the single-particle structure appears to be slightly expanded compared to the subtomogram averaging structure (Supplementary Video 3), perhaps due to S-layer curvature differences."

3. Are the pentameric defects reported in Fig. 4a, Extended Data Fig. 7 observable in tomograms or only predicted geometrically? The authors have previously reported a structure of similar pentameric csg defects in the S-layer of *Haloferax volcanii*. While a complete structure of pentameric Nmar_1547 may not be necessary, some analysis of these defects on a detailed structural level would be appropriate.

We thank the reviewer for their critique, which have improved substantially our analyses of the cryo-ET data. The pentameric defects are always observed at the edges of the cell, now shown by the analysis of cellular subtomogram averaging positions in Fig. 1.

"Fig. 1| Molecular structure and assembly of the *N. maritimus* S-layer in intact cells.

a, A denoised tomographic slice through a *Nitrosopumilus maritimus* cell shows ultrastructural details of this marine archaeon (annotated). Inset shows top and side views of the subtomogram average of

the S-layer. Scale bar: 500 Å; Inset scale bar: 100 Å. **b**, Map of the subtomogram positions in the cellular S-layer showing presence of pentameric defects at the edge of the cell. Scale bar: 500 Å. **c**, A histogram of subtomogram positions from all tomograms relative to the three-dimensional centre of the cell body (grey: hexamers, pink: pentamers).”

4. The authors report structures of the S-layer in *N. maritimus* consisting of Nmar_1547 and not Nmar_1201. While they do explicitly state that they cannot rule out the presence of low copy number Nmar_1201, one is left wondering whether this could be confirmed through 3D classification, especially considering the high proportion of picked particles not used for the final reconstruction with SPA.

We were extremely concerned about this, and therefore we had performed extensive 3D classification, but never observed any map that would be consistent with Nmar_1201 being the SLP. The large number of discarded particles arise because initially we over-sampled the data during picking and then removed particles that corresponded to the same hexamer (or aligned to the same hexamer). We have mentioned this in the methods section -

“Particles belonging to class averages centred at the (pseudo)-hexameric axis were combined, and particles within 30 Å were removed to prevent duplication after alignment.”

5. Line 99 in the manuscript refers to Figure S2 which purportedly shows cation densities near residues D73 and E74 in the C2 pore, but I could not find this figure. Additionally, extended Data Fig. 6d shows the residues at the pore, but the two residues mentioned in the text D73 and E74 do not appear to be in the pore region as shown here. Additionally, one would expect to also see these cation densities in Extended Data fig. 2 (or elsewhere).

Sorry about that; we have now pointed to these densities in Extended Data Fig. 2. However, reviewer 3 advised caution against interpretation of these densities; therefore, we have toned down the discussion on the densities in the text. Instead, we have provided direct biochemical evidence of ammonium binding.

“...we hypothesised that these additional densities on the *N. maritimus* S-layer could potentially correspond to bound cations, although the resolution of the map prevented us from unambiguously assigning their chemical identities.”

We also provide a detailed analysis of ammonium binding per residue through further MD simulations in Extended Data Figure 7.

“l, Occupancy profile for the modelled NmSLP sequence; residues binding to ammonium ions (high occupancy) are shown as large letters.”

6. Supplementary Video 1 shows a representative tomogram with a nicely segmented S-layer. Additionally nice would be a supplementary video of a representative tomogram without segmentation so the reader can appreciate the structure of the S-layer as seen directly in the tomogram.

We have provided this as requested, thank you for your comment.

“Supplementary Video 2| Cryo-electron tomography of *N. maritimus*.

Slices through tomogram of the same *N. maritimus* cell shown in supplementary video 1 without segmentation to enable direct assessment of the cryoET density.”

7. Has the tomographic slice in Fig 1a been subject to modification eg. Denoising? The background of the tomogram in this figure differs from that shown in Supplementary Video 1. If so, this should be reported in the Methods where appropriate.

Indeed, additional denoising was performed on this tomogram, now mentioned in the legend to Fig. 1.

“a, A denoised tomographic slice through a *Nitrosopumilus maritimus* cell shows ultrastructural details of this marine archaeon (annotated).”

and

“Tomograms for visualisation were generated using the simultaneous iterative reconstruction technique (SIRT) implemented in Tomo3D⁴³ and denoised with Cryo-CARE^{44,45}.”

8. The methods section for subtomogram averaging mentions the usage of custom scripts. The authors should include what these scripts were used for, and whether they are publicly available.

These scripts for initial subtomogram averaging are part of the AV3 package developed by Briggs and colleagues, and are publicly available from the original authors' website, mentioned in the cited manuscript. The link to download the scripts is provided below - https://www.biochem.mpg.de/7939859/Wan-et-al_-2017

9. The method by which particle positions for subtomogram averaging were attained is not mentioned in the methods (unless it is part of the custom scripts referred to in comment 8).

Yes, this was also part of the custom scripts, all available in the link above, or further detailed in the citations within the manuscript.

Referee #3 (Remarks to the Author):

Kügelgen et al. present a comprehensive approach in cryo-EM, cryo-ET, bioinformatic analysis, simulations and growth experiments to characterize membrane-less channels from the ammonia-oxidizing marine archaea *Nitrosopumilus maritimus*. In their study, the authors reveal the architectures of a molecular machinery that captures and directs ammonium ions to the cell membrane. The subtomogram with a resolution of 3.3 Å is very impressive, and the 2.7 Å resolution of the single particle electron cryomicroscopy reconstruction shows in detail how the S-layer is arranged to trap ammonia ions. In summary, the authors' analysis defines the power of tomography as a technology of the future, paving a new path in structural biology. The manuscript is well written. However, some terms are used incorrectly. References are sufficient, but SI figures need improvement to support the current manuscript. I recommend publishing the manuscript in Nature if my concerns can be addressed.

Thank you very much for your comments. We present a substantially revised manuscript, which we hope that you will like.

1) Line 25. Many others and I always have trouble defining an atomic structure with a resolution of about 3 Å. At atomic resolution, the atoms are separated from each other, or at least water molecules or the rings of the cyclic side chains can be seen. I would change that term to "high resolution", which is a more general term and covers this range of resolutions.

We agree, and we have changed this throughout the manuscript.

"Molecular structure determination of the *N. maritimus* S-layer from cells"
and

"..which have been shown to support high-resolution *in situ* structure determination from purified specimens¹⁹"

2) Line 26. In my opinion, *de novo* always describes a new fold of a protein and it's clearly not an entirely new type of structure but an interesting novelty. Especially since there is a solved structure (pdb id 7o23) with low id compared to Nmar_1547 showing similar folding in some areas. In addition, it has been shown in many examples that S-layers are assembled in pseudo-hexagonal arrays. Therefore, the authors should elaborate the novel properties of this complex in order to bring a better understanding to the reader.

Sorry about the confusion; we have removed the term *de novo*, which we were only using to show that direct model building was performed on the cellular cryoET density.

3) Line 30. Fix statement: SPA cryo-EM does not provide arrangement information of the S-layer in a lattice, but cryo-ET does.

We have changed the sentence as suggested.

"We supplemented our *in situ* structure of the ammonium-binding S-layer array with a 2.7 Å-resolution single-particle electron cryomicroscopy structure, revealing detailed features of this immunoglobulin-rich and glycan-decorated S-layer."

4) Line 42. 1028 cells sound like a lot, but the number is difficult for the reader to catch. Is there another order of magnitude that gives a better idea? Maybe how many tons of Amomium are converted or what proportion of these microorganisms have compared to others? As far as is known, sequencing shows that *Nitrosopumilus maritimus* make up about 1% of the metagenome on the surface of the oceans (Walker et al. PNAS 2010).

We agree with the reviewer and have removed this estimate from the manuscript text.

“The numerical dominance of marine thaumarchaea suggests that they have a major role in global biogeochemical cycles^{2,3}”

5) Extended Data Fig. 1a&b. The FSC curve clearly shows that the STA map has a resolution of 3.3 Å. In contrast, the Resmap has the best local resolution at around 3.2 Å. That concludes for me the overall resolution here is 0.5 Angstrom higher. And why does the scale bar range from 3.2 to 5 Å? Please clarify.

Sorry about the confusion; this was also a comment from reviewer 2. The resolution of the map ranges from ~3.3 Å in the centre of the hexamer to ~4.5 Å at the edge of the hexamer. This has now been clarified in the text throughout as well as in the Extended Data Fig. 1 (showing local resolution) and the methods section.

“Relaxation of the symmetry^{42,43} led to improved (3.3 Å) resolution overall, and 3.2 Å at the pseudo-hexameric axis, but decreased resolution (~4.5 Å) at the periphery of the hexamer (see Extended Data Table 1 and Extended Data Fig. 1).”

6) Line 70. A paper - focusing on structural studies should elaborate on "computational algorithms" and thus point towards a processing scheme at this point - this is clearly missing in the manuscript

This is an excellent critique. We have added our processing scheme to Extended Data Figure 1.

7) Line 75: pseudo-hexagonal - please add deviation from ideal C6 point group; was symmetry applied during processing?

Yes, C2 symmetry was applied final reconstruction after symmetry relaxation from C6. Mentioned in the Methods section now explicitly.

“...subtomogram improvements and realignments increased the resolution of the NmSLP hexamer to 3.4 Å in C6 symmetry. Relaxation of the symmetry^{42,43} led to improved (3.3 Å) resolution overall, and 3.2 Å at the pseudo-hexameric axis, but decreased resolution (~4.5 Å) at the periphery of the hexamer (see Extended Data Table 1 and Extended Data Fig. 1).”

8) Line 80. Local map quality in Extended Data Fig. 2. can hardly be assessed by the areas selected - smaller areas showing all side chains would be preferable. In addition, the authors present some side chains in the STA maps, so it would be nice to show the same for the SPA map.

This is a good point. We have added as requested in the new Extended Data Figure 4.

“Extended Data Fig. 4| Comparison of the STA and SPA *N. maritimus* S-layer reconstruction.

a, Close-up view of the STA map with the built NmSLP model (domain 1) showing resolved large and small side chains. **b**, Close-up view of the SPA map in the same view as shown in panel (a). **c-d**, Close-up view of the STA map (c) and (d) of domain 3 of the NmSLP with resolved small and bulky side chains. **e-f**, Close-up views of the N-glycan densities of the STA map (a) and SPA map (f) in domain 1. **g-h**, The density of the N-glycans is enhanced in the deepEMhanced⁶⁷ map of the same region. **i-l**, Close-up views of some of the N-glycan densities of the STA (i,k) and SPA map (j,l) of domain 4.”

9) Line 84. The map shows a preferred orientation with local anisotropy, which explains the peripheral drop in resolution, which is also reflected in the 3D Fourier shell correlation histogram. I suspect there is a lack of side views of the particles. Therefore, would it be possible to represent the angular distribution of the particles for the STA map as for the SPA map?

We have added this in the Extended Data Figure 1.

g Particle distribution

“g, Particle distribution from the 0° projection image.”

10) Line 94. Unexplained densities are not mentioned in Extended Data Fig. 2 - point towards the areas of interest.

Done as requested.

11) Line 101. Include region in Extended Data Fig. 2; add a sentence of caution: at the reported local resolution of about 3.2 Å, ions are barely visible. So, I would be very careful showing that in Fig 3a.

We agree, and this also raised by reviewer 2. We have added a sentence to clarify.

“...we hypothesised that these additional densities on the *N. maritimus* S-layer could potentially correspond to bound cations, although the resolution of the map prevented us from unambiguously assigning their chemical identities.”

We realise that this part of the manuscript was not optimally presented. We now show all the pores in the S-layer in Fig. 2, and carefully comment on the visibility of the ions. Rather than only relying solely on the observation of these densities, we present new biochemical data showing ammonium binding, MD analyses of ammonium binding and bioinformatics of AOA charge distributions to support our data.

12) Line 112. FSC curves with different point group symmetry applied are indicated, C1 reconstruction is missing. Please add.

Added as requested to Extended Data Fig. 3.

13) Line 113. Proof of "excellent agreement" by model-superposition. RMSD of both models required and overlay of maps - side-by-side comparison of different figures is not enough.

We have performed this analysis and reported the result in the manuscript text.

“The single-particle structure was very similar (RMSD 2.21 Å for the full composite model, 0.54 Å for residues 35-455 (refined in C2) and 1.34 Å for residues 466-1498 (refined in C6)) with the subtomogram averaging structure (Fig. 2, Extended Data Fig. 4 and Supplementary Video 3),...”

14) Line 118. Please extend your observation of missing residues by a possible explanation, as to why this area was not resolved; might not even be a methodical limitation?

We believe that this domain may be flexible compared to the rest of the S-layer. We comment on this in the manuscript now.

“The last Ig-like domain remains unresolved in our map, with only disordered, diffuse density observed beyond the ninth Ig-like domain of NmSLP in the pseudo-periplasmic space, indicating flexibility relative to the rigid part of the S-layer.”

15) Line 139. What kind of glycans are present? Density is in SPA data is of sufficient quality to build initial glycan trees, at least in Fig. 2b.

Unfortunately, because the chemical formula of these glycans is not known, model building into the map was not possible, despite our best attempts. There are a multitude of hexoses with different functional groups present in prokaryotic glycans, which we cannot distinguish between at this resolution. We comment on this in the manuscript text.

“While these densities do not support direct derivation of the chemical structure of the glycans, they clearly project away from the cell membrane...”

16) Line 150. Growth curves should be shown in color to better distinguish them.

Done as requested.

“Extended Data Fig. 5| Ammonium growth, ITC and cryo-EM at increased ammonium concentration.

a, Growth curves of *N. maritimus* in differing ammonium concentrations. Different curves are shown in different colour (colour bar is shown on the left)”

17) Line 160 & 677. Range of APBS result for surface electrostatic potential not suitable for good representation of smaller changes in charge distribution - would be better suited to show the solvent-accessible region within the “the membrane proximal C-terminus highly negatively charged” - charge over residues plot not really fitting to what you want to communicate with your data.

Thank you for this suggestion; we have clarified in the revised manuscript, with more detailed comparison with the MD data provided in Figure 4a-b. Please see our response to reviewer 2 above.

18) Line 178. Please add a reference to published data or your own data – missing.

We have added this as requested to the Extended Data Fig. 8.

“Extended Data Fig. 8| Ions in MD simulations and S-layer charges.

a-c, Normalised and averaged residence of ions across the S-layer in MD simulations, from the membrane proximal side (left) to the extracellular side (right).”

Discussion:

19) Add conclusion about importance of glycosylation - clearly not on all domains, explain inhomogeneous distribution towards the center of the structure.

We have added a comment as requested.

“The mesh-like arrangement of the glycans likely provides protection, potentially shielding the cell from phages²⁷. It might also enhance the hydrophilicity of the cell surface, making it suitable for marine environments.”

20) Add a theorem about the cone-like structure of each S-sheet hexamer and the implications of such a geometry to allow dense formation of the sheet around a curved surface (the cell).

We have added a comment as requested.

“The dome-shaped structure of the S-layer hexamer probably supports flexibility, allowing NmSLP to coat different parts of the cell membrane with varied curvature.”

We have also performed more analysis about the pentamer positions in the structure, enriching the discussion about the S-layer geometry on the cell. See our response to reviewer 2 above.

Reviewer Reports on the First Revision:

Referees' comments:

Referee #1 (Remarks to the Author):

The authors have addressed my comments. I support publication of the revised manuscript.

Referee #2 (Remarks to the Author):

I am satisfied with the responses of the authors and the revisions made. I still can't comment on the MD simulations though. Unfortunately the manuscript as is still shows that there was no additional information gained by doing subtomogram averaging other than identifying the few pentameric particles and mapping them back to the tomogram.

It's nice to hear from the authors that the organization of the S-layer in *N. maritimus* and *H. volcanii* are different and that there is little sequence similarity between the Ig-like domains of each. However the overall architecture and domain organization of the hexamers appears very similar and in the 2021 publication on *H. volcanii* the S-layer organization is presented in a way that appears very similar to what is reported here. Previously it was also shown that pentamers are found at areas of high membrane curvature for *H. volcanii* which is the major finding of the new cryoET analysis in *N. maritimus*. The authors should illustrate clearly how the reported S-layer organization is distinct from those previously reported. Perhaps, this is obvious to others more familiar with the field, but at first sight, these structures look almost identical.

All in all, I am not entirely convinced about how novel the findings are but maybe it's because I'm not familiar with the details of the archaeal S-layer field. But if the reviewers in the field are convinced regarding the novelty of this story also to the broad readership of Nature, then after the revisions there are at least no issues I find with the cryo-EM/ET analysis.

Minor comment:

It's nice to know what custom scripts they used. I would recommend this be reported and cited in the Methods as well.

Referee #3 (Remarks to the Author):

I worked intensively on the revised version over the holidays. I am not an expert in simulations, but all additions seem to be consistent and support the manuscript's statement about a molecular machinery that traps ammonium ions and directs them to the cell membrane. On this point, however, I rely on the opinion of my colleagues. Nevertheless, the authors have managed to address the points raised by the reviewers regarding cryo-EM, cryo-ET, bioinformatic analysis and growth

experiments. I recommend the manuscript for publication in Nature.

Referee #4 (Remarks to the Author):

I am going to comment only on the MD part - indeed the simulation protocol is valid and state-of-the-art. However, a single, 500ns long simulation is somewhat short and difficult to fully interpret, apart from a possible serendipitous agreement with experiments.

I'd recommend the authors to follow these guidelines:

<https://www.nature.com/articles/s42003-023-04653-0>

and

<https://pubs.acs.org/doi/10.1021/acs.jcim.3c00599>

In practice, error bars should be provided for all the values / curves derived from MD simulations. It is also not clear whether a full 500ns simulation was used to calculate the relative occupancies, or some period was disregarded (given that initially all NH₄⁺ ions are in the bulk probably certain part can be indeed disregarded). Any estimation of the simulation convergence would be also needed - for example, do the relative occupancies changes significantly in 50ns/100ns blocks? In extended fig 7 the authors provide "cation residence times (golden density)" without any scale? Finally, the authors mentioned that they perform MD simulations at different concentrations, but opted not to show a single data point from these simulations. I can't really understand this decision, there would be plenty of space in e.g. Fig 4 c, ext Fig 8 a-c. I am not going to push for a specific number of MD simulation copies right now, if the authors can provide reasonable metrics of convergence and error estimation, but if they would come up somewhat shaky, a few more simulations might be needed. I am aware that these changes would not, most likely, affect the conclusions of the study, however a publication in Nature should be held to the highest standards when it comes to statistics and data presentation. Especially the field of MD simulations has been plagued by bad statistics and lack of uncertainty estimation, coming even from very prominent groups (e.g. <https://elifesciences.org/articles/44718>) and it's time now to follow the best standards in the field. Before this is done, I cannot recommend the ms for publication.

Author Rebuttals to First Revision:

Response to Reviewers' Comments for von Kügelgen *et al*

Black text – original reviewers' comments

Blue text – our response

Red text – manuscript excerpt

We thank the reviewers for their valuable comments, which have helped us improve the manuscript.

Referee #1 (Remarks to the Author):

The authors have addressed my comments. I support publication of the revised manuscript.

Thank you.

Referee #2 (Remarks to the Author):

I am satisfied with the responses of the authors and the revisions made. I still can't comment on the MD simulations though. Unfortunately the manuscript as is still shows that there was no additional information gained by doing subtomogram averaging other than identifying the few pentameric particles and mapping them back to the tomogram.

Reviewer 4 who is an MD expert confirmed that the MD simulations are state of the art. We have provided further clarifications to reviewer 4's comments, please see below. Subtomogram averaging (STA) provides much more information than just the location of pentamers. For example, STA shows that the cells are completely coated with an S-layer on all sides, STA confirms the overall arrangement of the S-layer seen in our single particle structure, as well as revealing how a curved lattice seen in *in situ* STA compares to a flat one from single particle analysis (Supplementary Video 3).

It's nice to hear from the authors that the organization of the S-layer in *N. maritimus* and *H. volcanii* are different and that there is little sequence similarity between the Ig-like domains of each. However the overall architecture and domain organization of the hexamers appears very similar and in the 2021 publication on *H. volcanii* the S-layer organization is presented in a way that appears very similar to what is reported here. Previously it was also shown that pentamers are found at areas of high membrane curvature for *H. volcanii* which is the major finding of the new cryoET analysis in *N. maritimus*. The authors should illustrate clearly how the reported S-layer organization is distinct from those previously reported. Perhaps, this is obvious to others more familiar with the field, but at first sight, these structures look almost identical.

There are several quantifiable differences between the *N. maritimus* and the *H. volcanii* S-layer. Some of these include (but are not limited to), the presence of a heavily glycosylated sandwich domain 4 in NmSLP, much higher number of domains (10 in NmSLP vs. 6 in *H. volcanii* csg), different glycosylation patterns, as well as different cell-anchoring mechanisms. We have explicitly stated this in the revised manuscript.

“Although these SLPs share some structural similarities, they diverge significantly at the sequence level, as well as at the overall organisational level containing different number of domains (Extended Data Figs. 5c), enabling them to assemble into unique two-dimensional sheets^{23,27}, each with distinctly different glycosylation patterns and cell-membrane anchoring mechanisms.”

All in all, I am not entirely convinced about how novel the findings are but maybe it's

because I'm not familiar with the details of the archaeal S-layer field. But if the reviewers in the field are convinced regarding the novelty of this story also to the broad readership of Nature, then after the revisions there are at least no issues I find with the cryo-EM/ET analysis.

Thank you for confirming that there are no issues with cryo-EM/ET.

Minor comment:

It's nice to know what custom scripts they used. I would recommend this be reported and cited in the Methods as well.

We have cited these in the methods.

“Sub-tomogram averaging was performed using custom scripts written in MATLAB, described in detail elsewhere^{38,46}.”

Referee #3 (Remarks to the Author):

I worked intensively on the revised version over the holidays. I am not an expert in simulations, but all additions seem to be consistent and support the manuscript's statement about a molecular machinery that traps ammonium ions and directs them to the cell membrane. On this point, however, I rely on the opinion of my colleagues. Nevertheless, the authors have managed to address the points raised by the reviewers regarding cryo-EM, cryo-ET, bioinformatic analysis and growth experiments. I recommend the manuscript for publication in Nature.

Thank you. We provide comments and clarification to the MD reviewer below.

Referee #4 (Remarks to the Author):

I am going to comment only on the MD part - indeed the simulation protocol is valid and state-of-the-art.

We thank the reviewer for their comments, and we fully agree with their authoritative critique. We have provided answers to all the queries below.

However, a single, 500ns long simulation is somewhat short and difficult to fully interpret, apart from a possible serendipitous agreement with experiments.

We agree with this. To rectify this situation, we have now run the 0.1 M NH₄⁺ simulation presented in main text Fig. 4 in triplicate (each 500 ns in duration). These expanded data confirm our reported observations and the agreement with experiments. Regarding the length of simulation, we now show in Extended Data Fig. 8 that the ionic position distributions settle within the first 100 ns, and remain stable for the remaining time in each simulation. We suggest therefore that 500 ns is a reasonable timescale for sampling the relatively dynamic ionic interactions presented here.

“Extended Data Fig. 8| Ionic distributions through the S-layer in MD simulations.

a-c, Averaged residence of ammonium (NH_4^+) ions through (in the directional orthogonal to) the S-layer in MD simulations, from the membrane proximal side (left) to the extracellular side (right) across different time windows showing convergence of the MD simulations.”

and

“**h**, Convergence of sodium ion residence in MD-simulation (one out of three replicates is shown).”

and

“**k**, Convergence of chloride ion residence in MD-simulations (one out of three replicates is shown).”

I'd recommend the authors to follow these guidelines:

<https://www.nature.com/articles/s42003-023-04653-0>

and

<https://pubs.acs.org/doi/10.1021/acs.jcim.3c00599>

We confirm that we are following the relevant guidelines for reporting the simulations presented in this study.

In practice, error bars should be provided for all the values / curves derived from MD simulations.

We agree and now provide error bars based on 3 x 500-ns simulations for the ionic positions in Fig. 4c and Extended Data Fig. 8 as well as the residue-based ammonium occupancies reported in Extended Data Table 4. These demonstrate that our original interpretation of the simulation results is robust and we are glad for the chance to confirm this.

“c, A histogram of negatively charged residues along the S-layer, overlaid on the ammonium ion positions in the 0.1 M NH_4^+ MD simulations (distance is calculated from the closest, membrane proximal amino acid residue in the S-layer structure). Error bars in the averaged ammonium ion residence from three independent MD simulations (averaged over the last 400 ns of each simulation) denote ± 1 standard deviation.”

and

“i, Averaged residence of sodium (Na^+) ions in three independent simulations shown in (g) averaged over the last 400 ns of each simulation. Errors bars denote ± 1 standard deviation.”

and

“I, Averaged residence of chloride (Cl⁻) ions in three independent simulations shown in (j) averaged over the last 400 ns of each simulation. Errors bars denote ±1 standard deviation.”

We would also like to highlight that we have now provided standard deviations in Extended Data Table 4 showing ammonium binding residues (excerpt of the table shown below).

“**Extended Data Table 4 | Ammonium binding residues predicted by MD.** Listed are residues with average occupancy >50% as computed by PyLipID over three 0.1 M NH₄⁺ NmSLP hexamer simulations.

#	Residue	Avg Occ	std dev	#	Residue	Avg Occ	std dev
1	ASP71	50.55	10.33	32	ASP918	75.53	3.91

It is also not clear whether a full 500ns simulation was used to calculate the relative occupancies, or some period was disregarded (given that initially all NH₄⁺ ions are in the bulk probably certain part can be indeed disregarded).

The first 100 ns of each simulation were disregarded prior to trajectory analysis in line with the convergence of the ionic position distributions (see also next point). We have now clarified this in the methods section.

“Unless otherwise indicated, analyses were performed after disregarding the first 100 ns of each simulation to ensure equilibrium sampling.”

Any estimation of the simulation convergence would be also needed - for example, do the relative occupancies changes significantly in 50ns/100ns blocks?

We thank the reviewer for raising this point. In Extended Data Fig. 8, we now provide plots of the ammonium position distributions averaged over 100-ns blocks for the three 0.1 M NH₄⁺ simulations, demonstrating that the distributions settle into a stable pattern within 100 ns. Also shown above.

a
and

b
and

c
“a-c, Averaged residence of ammonium (NH_4^+) ions through (directional orthogonal to) the S-layer in MD simulations, from the membrane proximal side (left) to the extracellular side (right) across different time windows showing convergence of the MD simulations.”

Additionally, we provide similar plots of the Na^+ and Cl^- position distributions for one of the replicas, showing that these distributions settle even more quickly.

“h, Convergence of sodium ion residence in MD-simulation (one out of three replicates is shown).”

and

“k, Convergence of chloride ion residence in MD-simulation (one out of three replicates is shown).”

As a separate measure of convergence, we compared the list of high-occupancy (>50%) residues identified by PyLipID for each of the three 0.1 M NH_4^+ simulations (disregarding first 100 ns). This analysis highlighted 63, 65, and 63 residues respectively with a total of 55 residues shared between all three lists, confirming that similar binding distributions are being sampled between the simulations.

In extended fig 7 the authors provide "cation residence times (golden density)" without any scale?

Sorry about the confusion. This density shows ammonium occupancy in a single MD simulation (0.1 M ammonium), therefore there is no colour scale bar in this figure panel. We have clarified this in the figure legend now.

"b, Ammonium ion densities (ammonium occupancy during a single simulation shown as golden density) plotted onto the structure shown in ribbon representation. For further details on ammonium binding residues, please see Extended Data Table 4 and Extended Data Figure 9"

We would like to highlight that we also have Supplementary Video 4 with an appropriate colour scale bar for ammonium occupancy.

"Supplementary Video 4| Ammonium binding in MD simulations.

Residues identified by PyLipID with average ammonium occupancy >50% for three 0.1 M NH_4^+ NmSLP hexamer simulations. Each residue is coloured by its occupancy value, which was averaged over the six NmSLP monomers, and mapped on to the hexamer structure. Residue numbers and occupancies with standard deviations are given in Extended Data Table 4."

Additionally, we would like to highlight that the Extended Data Table 4 now contains an explicit list of ammonium binding residues, from the triplicate of simulation with standard deviations.

Finally, the authors mentioned that they perform MD simulations at different concentrations, but opted not to show a single data point from these simulations. I can't really understand this decision, there would be plenty of space in e.g. Fig 4 c, ext Fig 8 a-c.

We agree and have now included the ammonium position distributions from these simulations in Extended Data Fig. 8, which show a similar overall binding pattern for all three concentrations.

"d, Overlay of averaged residence of ammonium (NH_4^+) ions in three independent simulations averaged over the last 400 ns of each simulation. Note that the first 100 ns of each simulation

is not included to allow for equilibration (see Methods). **e-f**, Averaged residence of ammonium (NH_4^+) ions through the S-layer in MD simulations at **e**, reduced ammonium concentrations (0.05 M) and **f**, increased ammonium concentrations (0.2 M)”

Please note that unlike the main simulation shown in Fig. 4, these simulations were not run in triplicate, because we did not find any major differences by altering the ammonium concentration. We have updated the text and Methods accordingly to clarify the comparison.

“To ensure that the observed binding pattern did not depend on the amount of ammonium present, we conducted further MD simulations of the S-layer system soaked in 0.05 M or 0.2 M ammonium, reproducing in both cases strong ammonium binding by the S-layer and relative accumulation of ammonium ions towards the cell-facing side of the S-layer (Extended Data Fig. 8).”

and

“To assess the robustness of the observed ammonium binding pattern, we further constructed hexamer systems containing ammonium at concentrations of 0.05 M (156 NH_4^+ ions) and 0.2 M (624 NH_4^+ ions) using an identical procedure, which were subjected to a single 500-ns production simulation.”

I am not going to push for a specific number of MD simulation copies right now, if the authors can provide reasonable metrics of convergence and error estimation, but if they would come up somewhat shaky, a few more simulations might be needed.

In light of the above additions, we confirm that the ion binding results are robust as indicated by agreement across triplicate simulations, as well as simulations at different ammonium concentrations. Also, the maintenance of ionic positions through the S-layer after the first 100 ns of each simulation shows that the system has converged.

I am aware that these changes would not, most likely, affect the conclusions of the study, however a publication in Nature should be held to the highest standards when it comes to statistics and data presentation. Especially the field of MD simulations has been plagued by bad statistics and lack of uncertainty estimation, coming even from very prominent groups (e.g. <https://elifesciences.org/articles/44718>) and it's time now to follow the best standards in the field.

We are thankful that the reviewer gave us a chance to show the robustness of our observations. As predicted by the reviewer, these changes did not affect the conclusions of the study.

Before this is done, I cannot recommend the ms for publication.

We hope that with our additions and clarifications, the reviewer will be convinced.

Please note that we have included additional changes to adhere to Nature's submission and reporting guidelines, a tracked version is included for your convenience.

Reviewer Reports on the Second Revision:

Referees' comments:

Referee #4 (Remarks to the Author):

The authors addressed all my comments, and included robust error estimations where it was needed, from additional simulations. I therefore fully recommend the manuscript for publication and would like to congratulate the authors for this amazing story and publication.